# TDP-43 maximizes nerve conduction velocity by repressing a cryptic exon for paranodal junction assembly in Schwann cells

Kae-Jiun Chang[1], Ira Agrawal[2], Anna Vainshtein[3], Wan Yun Ho[2], Wendy Xin[1], Greg Tucker-Kellogg[4], Keiichiro Susuki[5], Elior Peles[3], Shuo-Chien Ling[2,6,7]*, Jonah R Chan[1]*

[1]Department of Neurology, Weill Institute for Neurosciences, University of California, San Francisco, San Francisco, United States; [2]Department of Physiology, National University of Singapore, Singapore, Singapore; [3]Department of Molecular Cell Biology, Weizmann Institute of Science, Rehovot, Israel; [4]Department of Biological Sciences, and Computational Biology Programme, Faculty of Science, National University of Singapore, Singapore, Singapore; [5]Department of Neuroscience, Cell Biology, and Physiology, Boonshoft School of Medicine, Wright State University, Dayton, United States; [6]NUS Medicine Healthy Longevity Program, National University of Singapore, Singapore, Singapore; [7]Program in Neuroscience and Behavior Disorders, Duke-NUS Medical School, Singapore, Singapore

*For correspondence:
phsling@nus.edu.sg (S-CL);
Jonah.Chan@ucsf.edu (JRC)

Competing interests: The authors declare that no competing interests exist.

**Abstract** TDP-43 is extensively studied in neurons in physiological and pathological contexts. However, emerging evidence indicates that glial cells are also reliant on TDP-43 function. We demonstrate that deletion of TDP-43 in Schwann cells results in a dramatic delay in peripheral nerve conduction causing significant motor deficits in mice, which is directly attributed to the absence of paranodal axoglial junctions. By contrast, paranodes in the central nervous system are unaltered in oligodendrocytes lacking TDP-43. Mechanistically, TDP-43 binds directly to *Neurofascin* mRNA, encoding the cell adhesion molecule essential for paranode assembly and maintenance. Loss of TDP-43 triggers the retention of a previously unidentified cryptic exon, which targets *Neurofascin* mRNA for nonsense-mediated decay. Thus, TDP-43 is required for neurofascin expression, proper assembly and maintenance of paranodes, and rapid saltatory conduction. Our findings provide a framework and mechanism for how Schwann cell-autonomous dysfunction in nerve conduction is directly caused by TDP-43 loss-of-function.

## Introduction

TDP-43 (transactivation response DNA-binding protein of 43 kDa, encoded by *Tardbp*) is a nucleic acid-binding protein that regulates the processing of a wide range of coding and noncoding RNAs by directly binding to transcripts and/or interacting with key RNA-processing complexes (*Ederle and Dormann, 2017*; *Ling et al., 2013*). TDP-43 binds to more than 6000 RNA targets in the brain (*Polymenidou et al., 2011*; *Tollervey et al., 2011*)—roughly 30% of the total transcriptome—and modulates target gene expression by regulating mRNA stability, splicing, polyadenylation site selection, transport, and translation (*Ederle and Dormann, 2017*; *Ling et al., 2013*; *Rot et al., 2017*). As such, TDP-43 is a multifunctional master gene regulator with numerous potential targets as its downstream effectors. The detection of TDP-43-containing pathological aggregates in numerous neurodegenerative diseases—including amyotrophic lateral sclerosis (ALS) and

frontotemporal dementia—has resulted in the classification of a spectrum of disorders known as TDP-43 proteinopathies (*Kwong et al., 2008*; *Ling et al., 2013*). It is widely considered that both loss of function and gain of toxic properties may contribute to the neurodegeneration associated with TDP-43 proteinopathies (*Ling et al., 2013*). Therefore, it is of critical importance to gain a better understanding of the physiological functions for TDP-43.

Not surprisingly, the function of TDP-43 is mainly focused in neurons (*Iguchi et al., 2013*; *Klim et al., 2019*; *Melamed et al., 2019*; *Tann et al., 2019*; *Wu et al., 2012*). However, emerging evidence indicates that the functional roles for TDP-43 are not limited to neurons, but that glial cells are also reliant on TDP-43 function and may play a role in pathogenesis (*Paolicelli et al., 2017*; *Peng et al., 2020*; *Velebit et al., 2020*; *Wang et al., 2018*). Intriguingly, it is known that the cells of the peripheral nervous system (PNS) are also affected in ALS patients with differential vulnerability (*Gentile et al., 2019*), calling for a better understanding of a potential cell-autonomous role for TDP-43 in various PNS cell types. Therefore, we decided to investigate the physiological role of TDP-43 in the main glial cell type of the PNS, namely the Schwann cell (*Stierli et al., 2018*). Schwann cells ensheathe and myelinate all relevant axons of the peripheral nerves (*Nave and Werner, 2014*; *Salzer, 2015*). Moreover, Schwann cell–axon interactions highly cluster voltage-gated sodium (Nav) channels at nodes of Ranvier, thus enabling rapid saltatory conduction (*Poliak and Peles, 2003*; *Salzer, 2003*). The intricate interactions between axons and Schwann cells further establish the paranodal axoglial junctions that flank nodes of Ranvier and are essential for the maximal insulatory function of myelin (*Pedraza et al., 2001*; *Rosenbluth, 2009*). Given the pivotal role Schwann cells play in the physiology of the PNS, we investigated the function of TDP-43 in Schwann cells. In mice lacking TDP-43 in Schwann cells, we find a 50% delay in nerve conduction with motor deficits, without any apparent structural alteration in compact myelin. Further analysis reveals a specific disruption in the assembly of paranodal junctions, directly resulting in conduction delay. This deficit is exclusive to Schwann cells and not observed in oligodendrocyte paranodes in the central nervous system (CNS). Furthermore, we find that TDP-43 is required for the expression of *Neurofascin* (*Nfasc*), which encodes a glial cell adhesion molecule necessary for paranodal junction formation and maintenance, by repressing the usage of a cryptic exon during splicing. Our findings are the first demonstration of a functional role for TDP-43 in axon–glial interactions in the PNS and that the loss of function for TDP-43 in Schwann cells results in impaired conduction velocity and motor behavior.

## Results

### Schwann cell TDP-43 is required for rapid nerve conduction but not for myelination

TDP-43 is abundantly expressed by all Sox10-positive Schwann cells (*Figure 1A*, *Figure 1—figure supplement 1A, B*). To elucidate the PNS-autonomous function of TDP-43, we specifically ablated TDP-43 from Schwann cells by combining the TDP-43 conditional allele (*Tardbp*$^{fl/fl}$) (*Chiang et al., 2010*) with *Dhh-Cre* (*Jaegle et al., 2003*). In these conditional knockout (cKO) mice, TDP-43 expression is completely abolished in Schwann cells (*Figure 1A*, *Figure 1—figure supplement 1*). By measuring the motor nerve conduction velocity, we find that action potential propagation is delayed by ~50% in the cKO mice at postnatal day (P) 27 (*Figure 1A, C*). However, the cKO sciatic nerves appear normal in size and opacity compared to wild-type (WT) nerves (*Figure 1D*), suggesting that compact myelin is being formed normally. To confirm the extent of myelination, we examined the sciatic nerves by electron microscopy. While a slight decrease in the number of myelinated axons is observed in the cKO nerves at P3, they recover to control levels by P21 (*Figure 1E, F*) with myelin thickness comparable to the WT (*Figure 1—figure supplement 2A*). Taken together, these results demonstrate that Schwann cell TDP-43 is required for normal nerve conduction velocity, despite the normal appearance of compact myelin in the TDP-43-cKO mice.

### Paranodal junctions are not formed in the PNS of TDP-43-cKO mice

If compact myelin is unaltered, then what mechanism underlies the conduction delay in the TDP-43-cKO nerves? Paranodal axoglial junctions serve as a multifunctional diffusion barrier that is required for maximal axon insulation provided by myelin sheaths (*Pedraza et al., 2001*; *Rosenbluth, 2009*). Mouse mutants lacking paranodal junctions display an ~40–50% delay in nerve conduction without

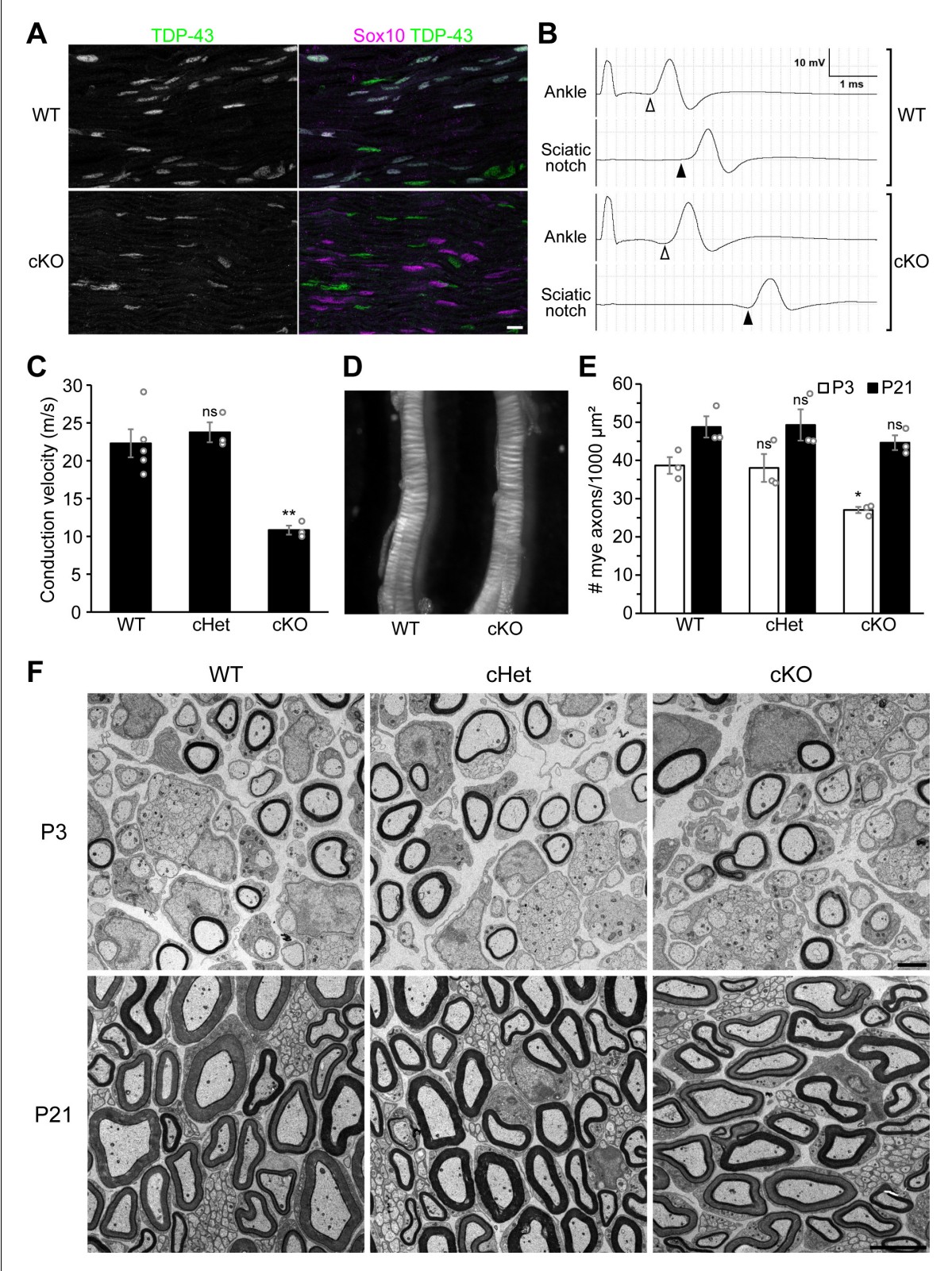

**Figure 1.** Knockout of TDP-43 in Schwann cells results in a 50% conduction delay without overt alteration of compact myelin. (**A**) Longitudinal sections of P28 wild-type (WT) and conditional knockout (cKO) sciatic nerves were immunostained for TDP-43 (green) and Sox10 (magenta). Sox10 labels Schwann cells, which are all TDP-43-negative in the cKO. The cell types other than Schwann cells are Sox10-negative and are still TDP-43-positive in the cKO. Scale bar, 10 μm. (**B, C**) Motor nerve conduction of P27 mice was measured as compound muscle action potentials at the plantar muscles evoked
*Figure 1 continued on next page*

*Figure 1 continued*

by stimulation of the nerve at the ankle and sciatic notch. The onset of the compound muscle action potentials is indicated by open arrowheads (ankle stimulation) and solid arrowheads (sciatic notch stimulation) in **B**. Bars represent mean ± SEM in **C**. n = 5 mice for WT, 3 for conditional heterozygote (cHet), and 3 for cKO. **p=0.0030 and 0.0028 (WT vs. cKO and cHet vs. cKO, respectively); ns: not significant, p=0.8094 (WT vs. cHet); one-way analysis of variance (ANOVA) with Tukey's test. (**D**) Sciatic nerves from P7 WT and cKO mice. (**E**) The number of myelinated axons per 1000 µm$^2$ was quantified with electron micrographs of sciatic nerve cross sections. Bars represent mean ± SEM. n = 3 mice per genotype. *p=0.039 and 0.048 (WT vs. cKO and cHet vs. cKO at P3, respectively); ns: not significant (WT vs. cHet at P3, p=0.9812; P21, p=0.5381); one-way ANOVA with Tukey's test. (**F**) Electron micrographs of P3 and P21 sciatic nerve cross sections. Scale bars, 2 µm for P3 and 5 µm for P21. cHet and cKO by *Dhh-Cre* (**A–F**).

The online version of this article includes the following source data and figure supplement(s) for figure 1:

**Source data 1.** Statistical summary for *Figure 1C, E* and *Figure 1—figure supplement 2*.

**Figure supplement 1.** TDP-43 is expressed by wild-type (WT) Schwann cells and specifically ablated from the conditional knockout (cKO) Schwann cells.

**Figure supplement 2.** Quantification for the myelin thickness and axon diameters of myelinated axons in the TDP-43-cKO sciatic nerves.

gross myelination defects (*Bhat et al., 2001*; *Pillai et al., 2009*; *Susuki et al., 2013*). Therefore, we tested the hypothesis that conduction delay in the TDP-43-cKO nerves may be due to a defect in paranodal junctions. Paranodal junctions comprise three key cell adhesion molecules—axonal Caspr (contactin-associated protein), axonal contactin (Cntn), and glial neurofascin (NFasc) 155 kDa isoform (NF155), each of which is essential for paranodal junction formation (*Poliak and Peles, 2003*; *Salzer, 2003*). In the mature WT PNS, Nav channels and axonal 186 kDa isoform of NFasc (NF186) are enriched at nodes of Ranvier, whereas NF155 and Caspr/Cntn are highly clustered at paranodal domains flanking each node (*Figure 2A, B* [nodes indicated by open arrowheads and paranodes by solid arrowheads], *Figure 2C*). In stark contrast to the WT and conditional heterozygote (cHet), the prototypical Caspr clustering is no longer detected at paranodes in the cKO at P28 (*Figure 2A*) and neither are Cntn, NF155, and the NF155-associated scaffold protein AnkB (*Chang et al., 2014*; *Figure 2B–D*), indicating that paranodal junctions are disrupted in the TDP-43 cKO. To determine whether paranodal junctions are not assembled during early development in the TDP-43 cKO or are initially formed but not stabilized during and after myelination, we examined P3 sciatic nerves (initiation of myelination) and found that Caspr is absent at paranodes while the Schwann cell-secreted extracellular matrix component gliomedin (Gldn) (*Eshed et al., 2005*) remains clustered at nascent nodes (*Figure 2E*). This strongly suggests that paranodal junctions fail to form starting from the initiation of myelination in the absence of Schwann cell TDP-43.

One hallmark alteration of losing the diffusion barrier function exerted by paranodal junctions is translocation of the *Shaker*-type voltage-gated potassium (Kv1) channels and their associated protein complex from juxtaparanodal regions to paranodes (*Poliak and Peles, 2003*; *Salzer, 2003*). In WT nerves, Kv1.1 and Kv1.2 channels and Caspr2 are concentrated in the juxtaparanodal domains and separated from nodes of Ranvier by paranodal junctions (*Figure 2F*, *Figure 2—figure supplement 1A–C*). Strikingly, in the cKO sciatic nerves these components invade the paranodes (*Figure 2F*, *Figure 2—figure supplement 1A–C*). By examining trigeminal nerves and using two alternative Cre-driver lines—*Mpz-Cre* for Schwann cells (*Feltri et al., 1999*; *Feltri et al., 2002*) and *Cnp-Cre* for both Schwann cells and oligodendrocytes (*Lappe-Siefke et al., 2003*)—we consistently identify the complete loss of paranodal junctions and translocation of the Kv1 complex to paranodes in the TDP-43 cKO (*Figure 2—figure supplement 1*). Ultrastructurally, the paranodal loops in the cKO detach from the axons. The loops are no longer tightly apposed to the axon, resulting in the absence of the clear indentation of the axolemma, typically observed in WT nerves (*Figure 2G*). Taken together, our findings strongly suggest a cell-autonomous role of Schwann cell TDP-43 in assembling paranodal junctions and properly establishing the polarized molecular domains at nodes of Ranvier.

Mutant mice lacking paranodal junctions in both the PNS and CNS exhibit tremors, paralysis, and ataxia (*Berglund et al., 1999*; *Bhat et al., 2001*; *Coetzee et al., 1996*; *Pillai et al., 2009*). Despite the lack of paranodal junctions in the PNS, the TDP-43-cKO mice (by *Dhh-Cre* or *Mpz-Cre*) do not exhibit a gross neurological deficit when observed in their home cage. However, when challenged with the rotarod test, the cKO mice perform significantly worse than WT littermates (*Figure 2H*), suggesting that PNS paranodal junctions are required for optimal motor performance.

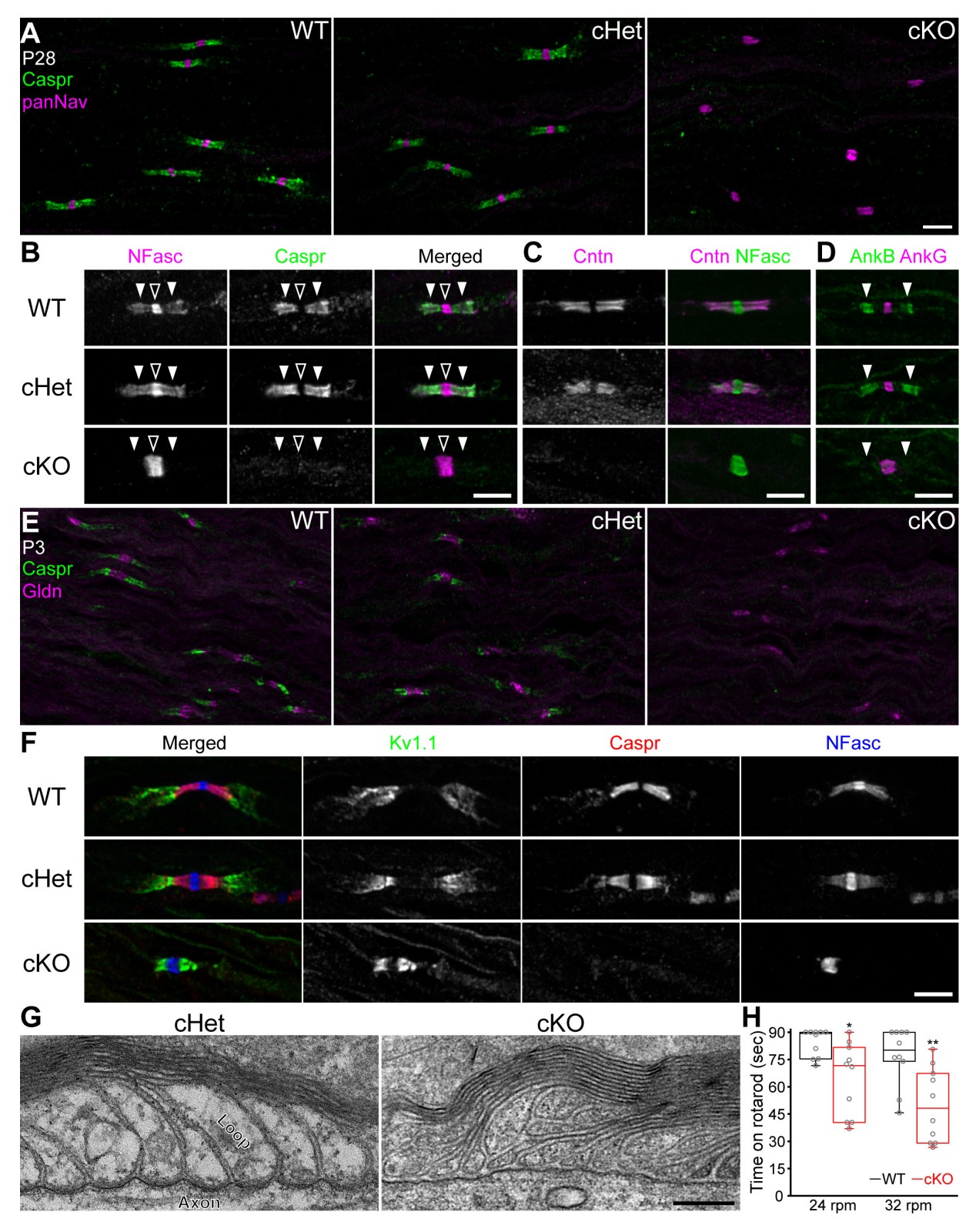

**Figure 2.** Paranodal axoglial junctions are disrupted in the TDP-43-cKO mice. (**A**) Immunostaining of P28 sciatic nerves for contactin-associated protein (Caspr) (green, paranode) and Nav channels (panNav, magenta, node). (**B–D**) Immunostaining of P28 sciatic nerves for (**B**) neurofascin (NFasc) (magenta, node and paranode) and Caspr (green), (**C**) contactin (Cntn, magenta, paranode) and NFasc (green), and (**D**) ankyrinB (AnkB, green, paranode) and ankyrinG (AnkG, magenta, node). Nodes are indicated by open arrowheads in **B**. Paranodes are indicated by solid arrowheads in **B, D**.
*Figure 2 continued on next page*

*Figure 2 continued*

(**E**) Immunostaining of P3 sciatic nerves for Caspr (green) and gliomedin (Gldn, magenta, node). (**F**) Immunostaining of P28 sciatic nerves for Kv1.1 channels (green, juxtaparanode), Caspr (red), and NFasc (blue). (**G**) Electron micrographs of P21 sciatic nerve longitudinal sections. (**H**) The time periods for which the mice remained on the rotarod at 24 or 32 rpm. Minima, first quartiles, medians, third quartiles, and maxima are plotted as box-and-whisker plots. n = 10 mice per genotype. *p=0.0115 for 24 rpm; **p=0.0036 for 32 rpm; Mann–Whitney tests. Scale bars, 5 µm (**A–F**) and 200 nm (**G**). Conditional heterozygote (cHet) and conditional knockout (cKO) by *Dhh-Cre* in (**A–C**) and (**E–G**), and by *Mpz-Cre* in (**D, H**).

The online version of this article includes the following source data and figure supplement(s) for figure 2:

**Source data 1.** Statistical summary for *Figure 2H*.

**Figure supplement 1.** Paranodal axoglial junctions are completely disrupted in the peripheral nervous system of the TDP-43-cKO mice using three different Cre-driver lines.

## Paranodal junctions are formed in the CNS of TDP-43-cKO mice

Given that paranodal junction assembly in both the PNS and CNS requires Caspr, Cntn, and NF155 (*Poliak and Peles, 2003*; *Salzer, 2003*), we set out to determine whether TDP-43 in oligodendrocytes is also essential for establishing axoglial junctions in the CNS. Although TDP-43 ablation in oligodendrocytes by *Cnp-Cre* causes necroptosis of mature oligodendrocytes (*Wang et al., 2018*), many myelin internodes exist in the CNS, which are formed by oligodendrocytes prior to degeneration. Surprisingly, Caspr clusters are still clearly detected in the cKO spinal cords (*Figure 3A, B*), indicating that TDP-43 is dispensable for CNS paranodal junction formation. The dramatic difference in the requirement for TDP-43 between the PNS and CNS is highlighted at the dorsal root transition zone, where the paranodal junctions fail to form exclusively on the PNS side in the cKO and not on the CNS side (*Figure 3A*, *Figure 3—figure supplement 1*).

## Fine myelin structures are preserved in the absence of TDP-43

Schwann cells are highly polarized cells, forming multiple structures to facilitate their functions (*Salzer, 2015*). In view of the potential perturbation in the expression of numerous TDP-43 targets in the cKO Schwann cells, it is likely that other structural changes could be identified, in addition to paranodal junctions.

Cajal bands are cytoplasmic-network conduits distributed outside compact myelin and are proposed to transport mRNA and proteins from the cell body (*Salzer, 2015*). We examined the nerves for βII spectrin (*Susuki et al., 2011*) and found that Cajal bands are clearly visible in the absence of TDP-43 (*Figure 4A*). Within compact myelin resides the inner tongue—the innermost cytoplasmic channel of a myelinating Schwann cell. At the interface between the axolemma and the tip of the inner tongue, axoglial interactions create a strand of internodal specialization winding around the axon, called the juxtamesaxon (*Poliak and Peles, 2003*; *Salzer, 2003*). The Kv1 complex is localized at the juxtamesaxon, forming a double-stranded structure (*Figure 4B*). In the absence of TDP-43, the juxtamesaxon is formed normally (*Figure 4B*). Additionally, the lateral cytoplasmic channels of myelinating Schwann cells wrap around the axon and form paranodal loops, which are interconnected by autotypic tight, gap, and adherens junctions (*Salzer, 2003*). By immunostaining for the junctional components (zonula occludens-1 [ZO-1], connexin 32 [Cx32], and E-cadherin [E-cad], respectively), we find that they are all normally enriched at paranodes in the cKO (*Figure 4C–E*). Lastly, Schmidt–Lanterman incisures, the cytoplasmic channels that run through the compact myelin and connect the outermost and innermost cytoplasmic compartments of Schwann cells, can be identified by staining for myelin-associated glycoprotein (MAG). Once again, the incisures were normally formed, as observed by the classic funnel-shaped MAG staining in the TDP-43 cKO (*Figure 4F*). Taken together, by examining the fine structures found in Schwann cell myelin, our findings suggest that the major effect of ablating TDP-43 in Schwann cells specifically lies in the disruption of paranodal junctions.

## NFasc expression requires TDP-43

How does TDP-43 deletion in Schwann cells induce the specific disruption of paranodal junctions? Among the known essential components, Caspr/Cntn and NF155, only NF155 is expressed by Schwann cells. Therefore, we asked whether the expression or the paranodal clustering of NF155 is impaired in the TDP-43 cKO. Consistent with disrupted expression of NF155, not only is NFasc

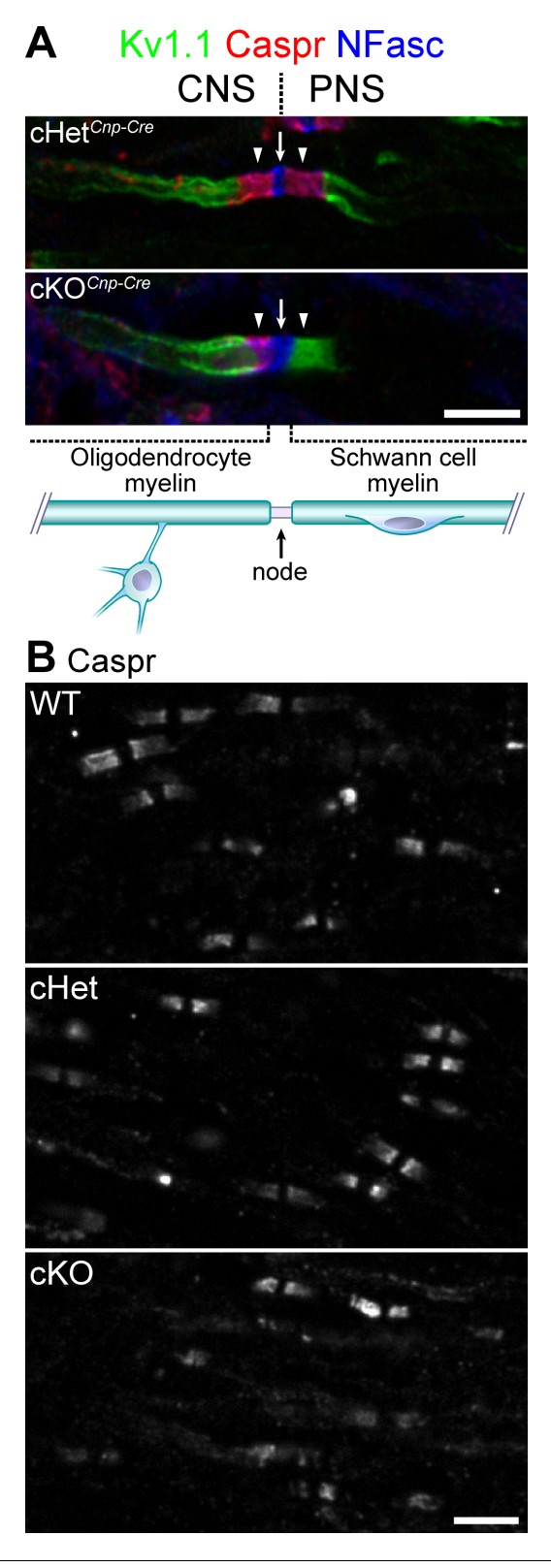

**Figure 3.** Knockout of TDP-43 in oligodendrocytes displays normal paranodal axoglial junctions in the central nervous system (CNS). (**A**) Immunostaining of the dorsal root entry zone from P60 conditional heterozygote (cHet) and conditional knockout (cKO) (by *Cnp-Cre*) spinal cords for Kv1.1 channels (green), contactin-associated protein (Caspr) (red), and neurofascin (NFasc) (blue). The node at the transition zone is indicated by an arrow and

*Figure 3 continued on next page*

*Figure 3 continued*

is shared by an oligodendrocyte at the left and a Schwann cell at the right, as illustrated below. The flanking CNS and peripheral nervous system (PNS) paranodes are indicated by arrowheads. Illustration adapted from Figure 1a of *Chang et al., 2016*, with permission. (B) Immunostaining of P21 spinal cords for Caspr. cHet and cKO by *Cnp-Cre*. Scale bars, 5 μm (A, B).

© 2016, Springer Nature. Figure 3A is adapted from Figure 1a of *Chang et al., 2016*, with permission from Springer Nature. Further reproduction of this panel would need permission from the copyright holder.

The online version of this article includes the following figure supplement(s) for figure 3:

**Figure supplement 1.** Nodes at the central nervous system–peripheral nervous system (CNS-PNS) transition zone are identified by immunostaining for the PNS paranodal marker ankyrinB (AnkB).

staining absent from the paranodes in the TDP-43 cKO (*Figure 2*, *Figure 2—figure supplement 1*), but NFasc is also undetectable at the incisures (*Figure 4F*). By contrast, although NF155 is not highly clustered at the paranodes in mice lacking Caspr or galactocerebroside/sulfatide, it remains localized at the incisures (*Poliak et al., 2001*; *Figure 5—figure supplement 1*). Taken together, these data strongly suggest that NF155 expression is abolished in the absence of TDP-43.

NF155 is encoded by *Nfasc,* a highly alternatively spliced gene (*Basak et al., 2007*; *Figure 5A*; *Supplementary file 1*). Alternative splicing generates NFasc proteins with different domain architectures of the extracellular moiety in different cell types (*Davis et al., 1996*; *Davis et al., 1993*; *Tait et al., 2000*). In Schwann cells, *Nfasc* mRNA contains exons 23 and 24 while exons 28–30 are excluded, producing the glial-specific NF155 isoform. In neurons, *Nfasc* transcripts are spliced in the opposite way and are translated into the neuronal-specific NF186 isoform. When the expression of NFasc proteins in sciatic nerves is analyzed by western blotting, we find that NF155 is undetectable in the TDP-43 cKO whereas the level of NF186 is unchanged (*Figure 5B*). Similarly, the protein expression of Caspr in the cKO nerve also remains unchanged (*Figure 5B*). Moreover, by immunostaining the nascent nodes and paranodes in P3 sciatic nerves, NFasc is already absent from the cKO paranodes (*Figure 5C*). Consistently, the mRNA for NF155 is dramatically decreased in the cKO (*Figure 5D*). Therefore, our findings strongly suggest that ablating TDP-43 in Schwann cells leads to aberrant NF155 expression and in turn results in the failure of Caspr to cluster at paranodes and form paranodal junctions.

In addition to NF155 and NF186, the other major NFasc isoform is NF140 (*Davis et al., 1993*), which contains neither exons 23–24 nor exons 28–30. Unlike NF186, NF140 protein is only faintly detectable in the TDP-43 cKO (*Figure 5B*). Using the isoform-specific primers in the RT-qPCR analysis, we find that NF140 containing exon 27 (NF140+ex27) remains unchanged, but NF140 without exon 27 (NF140-ex27) dramatically decreases in the TDP-43 cKO (*Figure 5D*), suggesting a neuronal source of NF140+ex27 (*Zhang et al., 2015*) and a glial source of NF140-ex27. Together, our observations point to a model where TDP-43 either directly or indirectly regulates the expression of NFasc in Schwann cells, which in turn interacts with Caspr/Cntn on the axolemma and assembles the paranodal junctions.

## TDP-43 directly regulates NFasc expression by repressing the inclusion of a cryptic exon

To investigate how TDP-43 regulates *Nfasc* expression, RNA-seq was performed using sciatic nerves isolated from the WT and cKO mice (by *Cnp-Cre*). Intriguingly, when the sequencing reads are aligned to the *Nfasc* locus and analyzed, we find reads mapped to introns 17 and 18 of *Nfasc*. Because intron 17 contains a UG-rich sequence, a known consensus RNA-binding motif for TDP-43 (*Polymenidou et al., 2011*; *Tollervey et al., 2011*; *Figure 6A, B*), we data-mined the previously published cross-linking immunoprecipitation and high-throughput sequencing (CLIP-seq) data on TDP-43-binding transcripts in mouse brains (*Polymenidou et al., 2011*) and identify a single cluster of pre-mRNA binding peaks on *Nfasc* located at intron 17 (upper panel of *Figure 6A*). Further analysis reveals a 104-bp-long cassette exon in the cKO with a well-defined splice acceptor and splice donor, just upstream to the TDP-43-binding peaks (*Figure 6B*), characteristic of TDP-43-repressed cryptic exons (*Ling et al., 2015*). We term this region as cryptic exon 1 (CE1). Quantitative splice junction analyses using the percent spliced in index (PSI) reveal a near complete utilization of the cryptic exon splice acceptor (93.1%) in the cKO sciatic nerves compared with 25.3% in the WT nerves

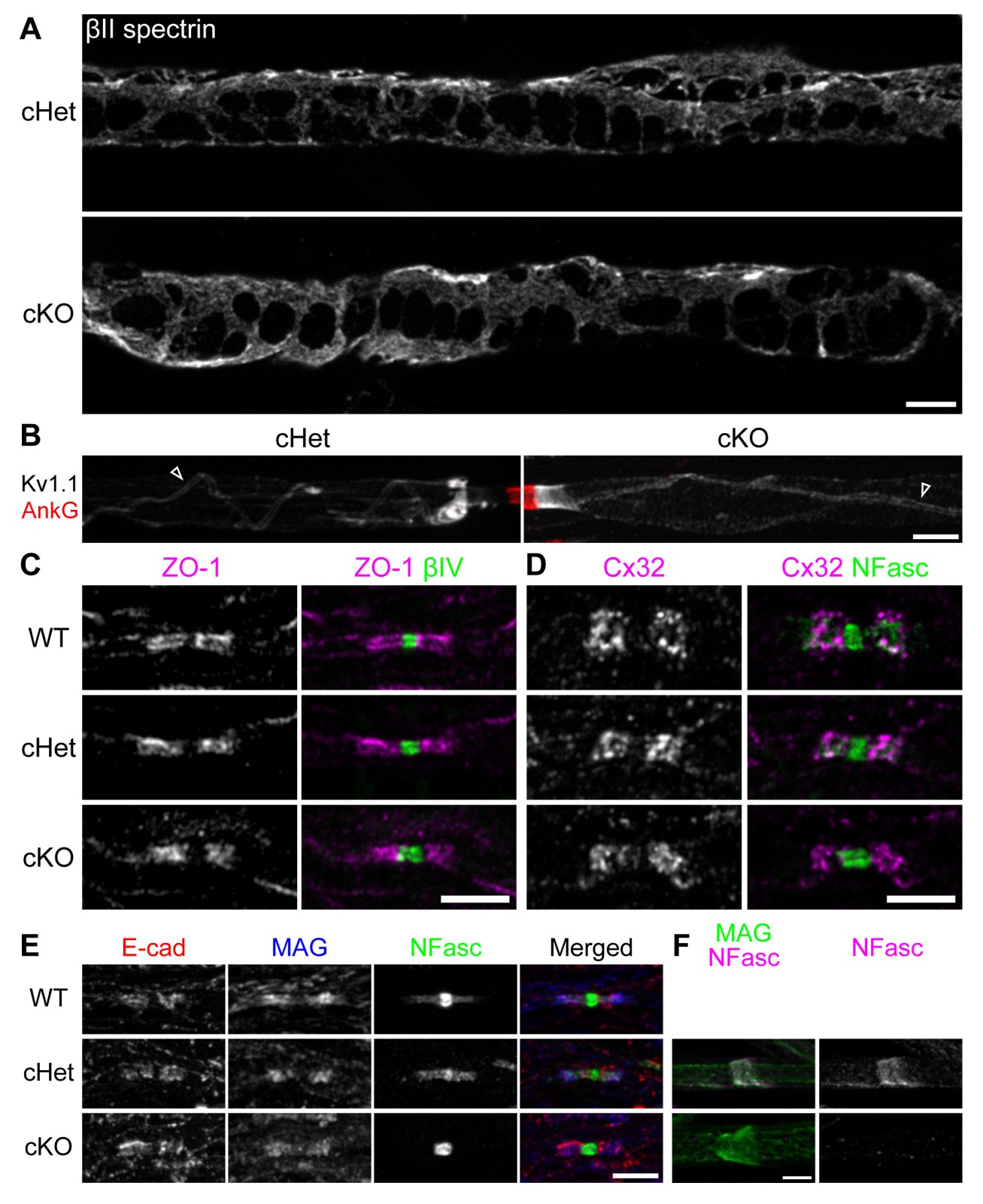

**Figure 4.** The fine myelin structures are formed normally in the TDP-43-cKO Schwann cells. (**A**) Immunostaining of Cajal bands in teased P29 ventral roots with an antibody against βII spectrin. (**B**) Immunostaining of teased P29 ventral roots for ankyrinG (AnkG) (red, node) and Kv1.1 channels (white). Localization of Kv1.1 channels in the juxtamesaxon is indicated by open arrowheads. (**C**) Immunostaining of P28 sciatic nerves for zonula occludens-1 (ZO-1) (magenta) and βIV spectrin (green, node). (**D**) Immunostaining of P28 trigeminal nerves for connexin 32 (Cx32) (magenta) and

*Figure 4 continued on next page*

*Figure 4 continued*

neurofascin (NFasc) (green). (**E**) Immunostaining of P28 sciatic nerves for E-cadherin (E-cad) (red), myelin-associated glycoprotein (MAG) (blue), and NFasc (green). (**F**) Immunostaining of teased P28 sciatic nerves for MAG (green, incisure) and NFasc (magenta). cHet and cKO by *Dhh-Cre* (**A–F**). Scale bars, 5 µm (**A–F**).

(*Figure 6—figure supplement 1*). By contrast, the usage of this cryptic splice acceptor is less dramatic (43.9%) in the spinal cords from the same TDP-43-cKO mice (*Figure 6—figure supplement 1*).

More critically, this newly identified cryptic exon region contains a premature stop codon when spliced with the upstream constitutive exon 17 (*Figure 6B*). A premature stop codon is the trigger for nonsense-mediated decay (NMD) (*Karousis and Mühlemann, 2019*), thereby providing the mechanistic explanation to the dramatic reduction of *Nfasc* transcripts and the absence of NFasc protein in the TDP-43-deleted Schwann cells (*Figure 4F*, *Figure 5*). A discrete band representing a potential truncated NFasc protein (with calculated molecular weight of 69 kDa before glycosylation, caused by the premature stop codon) is undetectable in the cKO by western blotting (*Figure 5B*), suggesting that the remaining 20% transcripts could be en route for degradation. Additionally, there are many less representative splice junction reads (<5 reads per species) in the region between CE1 and exon 19 (*Figure 6A*, *Figure 6—figure supplement 2*), suggesting weaker splice donor usage downstream of CE1. These reads are also increased upon TDP-43 deletion (*Figure 6A*, *Figure 6—figure supplement 2*), and we collectively term these cryptic exon extensions CE2 and CE3 (*Figure 6—figure supplements 1–3*). Regardless of the usage of CE1, CE2, or CE3, the premature stop codon is present in all these transcripts and thus render them substrates for NMD (*Figure 6—figure supplement 3C*).

To further confirm the presence of this cryptic exon in *Nfasc*, RT-PCR was performed. The predominant *Nfasc* isoform uses exons 17 and 19 and excludes exon 18 in the WT and cHet mice (*Figure 6C*). Consistent with the bioinformatic analysis, the predominant *Nfasc* isoform uses CE1 in the cKO sciatic nerves (*Figure 6C*). The precise splice junctions are confirmed by Sanger sequencing of the PCR products (*Figure 6Ci-iii*). By contrast, the utilization of CE1 is not observed in the majority of *Nfasc* transcripts in the spinal cords from the same TDP-43-cKO mice, and the CE1-containing PCR product is only faintly detected. Taken together, the data indicate an increased usage of the cryptic exon in *Nfasc* upon the loss of TDP-43 in the myelinating glia, where TDP-43 is required for proper splicing of the vast majority of *Nfasc* transcripts in Schwann cells, but not oligodendrocytes.

## TDP-43 is required for maintaining paranodal junctions in adult mice

Gene expression profiles change dramatically when Schwann cells transition from actively myelinating to the mature state (*Patzig et al., 2011*; *Siems et al., 2020*). Therefore, whether TDP-43 continues to control NFasc expression in mature Schwann cells after myelination needs to be addressed. It is clear from the analysis of the P60 cKO nerves that paranodal junctions are still absent (*Figure 2—figure supplement 1D, E*, *Figure 3A*), suggesting a lack of compensatory factors in the mature state that may bypass the need for TDP-43. To determine if loss of TDP-43 in mature Schwann cells results in termination of NFasc expression and then disassembly of paranodal junctions, we combined *Tardbp^{fl/fl}* with *Mpz-CreERT2* (*Leone et al., 2003*) (TDP-43 icKO) to conditionally induce deletion of TDP-43 in adulthood (~40–60% TDP-43 recombination efficiency in Krox20⁺ Schwann cells). At 3 months post-tamoxifen injection, NFasc is absent from a similar proportion of Schmidt–Lanterman incisures in the sciatic nerves and ventral roots (*Figure 7A*, *Figure 7—figure supplement 1A*). RT-PCR analysis of the sciatic nerves reveals a dramatic increase in the CE1 usage of *Nfasc* transcripts (*Figure 7B*). These observations suggest that TDP-43 continues to play an essential role in regulating NFasc expression by repressing the CE inclusion even in the mature state.

NFasc is notorious for its stability in the molecular domains where it is involved and requires a long time for protein turnover after adult or juvenile NFasc deletion (8–16 weeks at nodes; ≥8 weeks at paranodes) (*Desmazieres et al., 2014*; *Pillai et al., 2009*; *Zonta et al., 2011*). While NFasc protein is absent from the TDP-43-icKO incisures with residual NFasc still detected at paranodes to different extent, the paranodal diffusion barrier is compromised (*Figure 7C*, *Figure 7—figure supplement 1B*). This ranges from paranodal invasion of Kv1.1 to fragmentation or shortening of Caspr clusters (*Figure 7C*), and is consistent with the ongoing disassembly of paranodal junctions, similar to when *Nfasc* is ablated from mature Schwann cells (*Pillai et al., 2009*). The proportion of

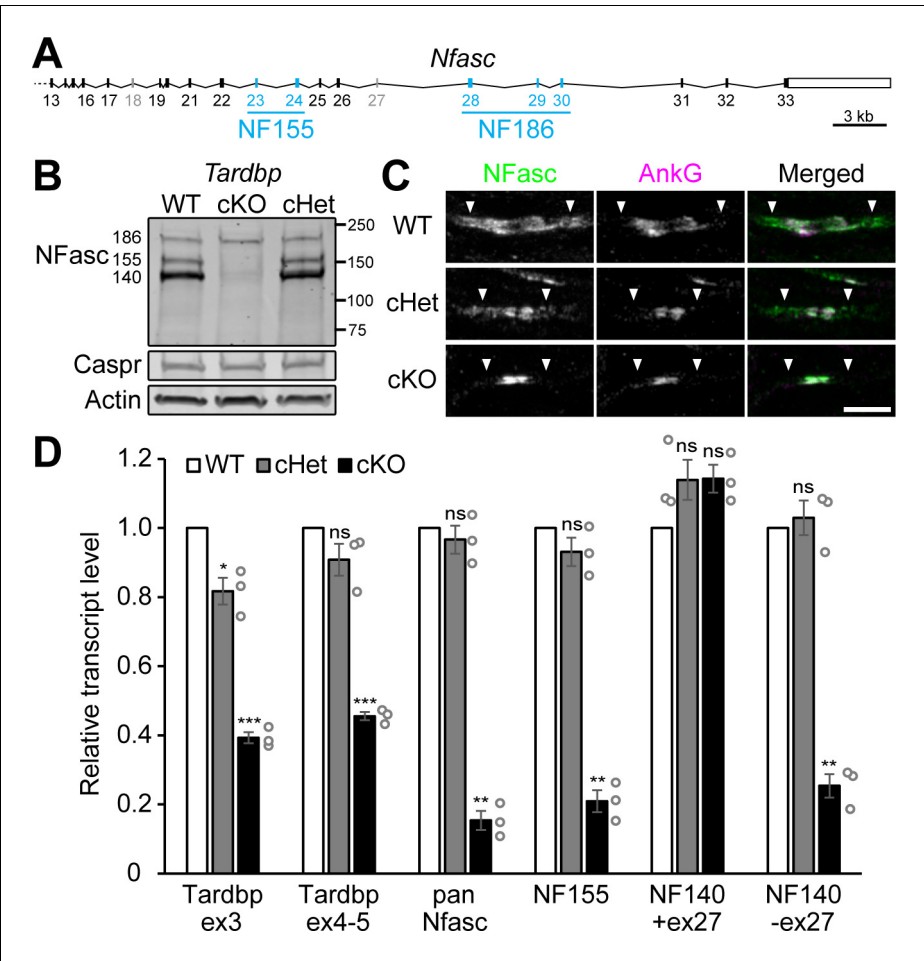

**Figure 5.** Schwann cell neurofascin (NFasc) expression is abolished in the TDP-43 conditional knockout (cKO). (**A**) The schema of mouse *Nfasc* locus showing the region from exon 13 to exon 33. The coding strand of *Nfasc* is located on the reverse strand of chromosome 1, as defined in the GRCm38/mm10 mouse genome assembly, and flipped here for reader convenience. Alternative exons specific to NF155 (exons 23 and 24) or NF186 (exons 28–30) are shown in blue while other alternative exons (exons 18 and 27) are shown in gray. NF140 contains neither exons 23 and 24 nor exons 28–30. The open rectangle denotes the untranslated region, and the solid ones the open reading frame. (**B**) Western blotting of P28 sciatic nerve homogenates probed for NFasc, contactin-associated protein (Caspr), and actin. (**C**) Immunostaining of P3 sciatic nerves for NFasc (green) and ankyrinG (AnkG) (magenta) showing pairs of approaching heminodes (nodes flanked by myelin sheaths only on one side). The paranodes are indicated by arrowheads. Scale bar, 5 μm. (**D**) RT-qPCR analysis of P28–P29 sciatic nerves shows the transcript levels relative to those of the wild-type (WT) after normalized to the internal control *Polr2a*. Exon 3 of *Tardbp* is the floxed exon, and the primer pair for *Tardbp* ex3 detects TDP-43 mRNA transcribed from the unrecombined allele. The primer pair for *Tardbp* exons 4–5 detects the nonfloxed region to confirm mRNA degradation after Cre recombination. Pan-*Nfasc* is detected by primers spanning exons 12–13, NF155 by primers spanning exons 22–23, NF140+ex27 by primers annealing to the junctions of exons 22/25 and exons 27/31, and NF140-ex27 by primers annealing to the junctions of exons 22/25 and exons 26/31. Bars represent mean ± SEM. n = 3 mice per genotype. *p<0.05; **p<0.01; ***p<0.001; ns, p≥0.05; one-sample unpaired two-tailed *t*-tests (WT vs. conditional heterozygote [cHet] and WT vs. cKO). cHet and cKO by *Dhh-Cre* (**B–D**).

The online version of this article includes the following source data and figure supplement(s) for figure 5:

**Source data 1.** Statistical summary for *Figure 5D*.

**Figure supplement 1.** Localization of neurofascin (NFasc) in the Schmidt–Lanterman incisure is not affected in the contactin-associated protein knockout (Caspr KO).

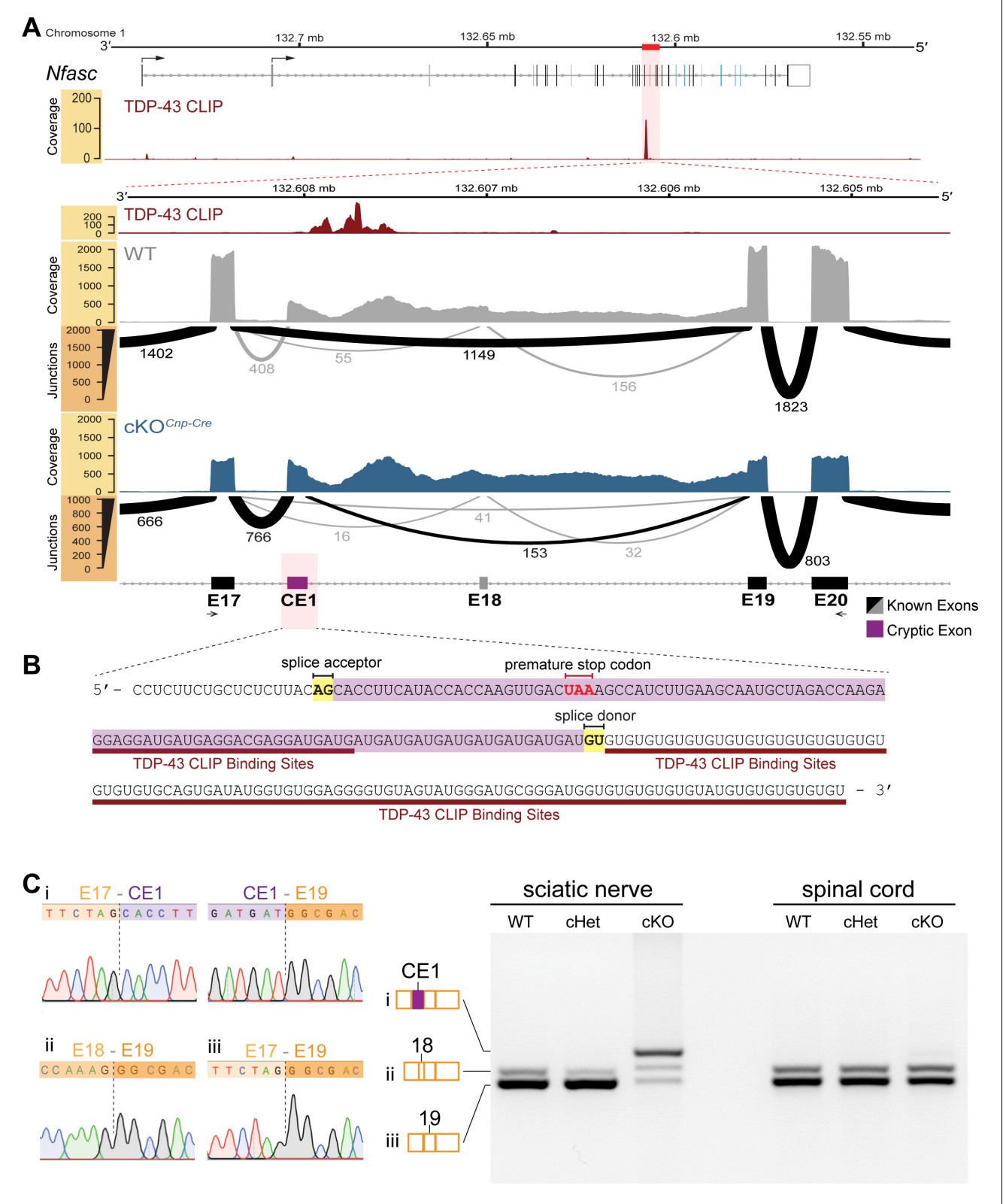

**Figure 6.** Cryptic exon located upstream of TDP-43-binding sites on *Nfasc* mRNA is highly expressed in the TDP-43 conditional knockout (cKO). (**A**) TDP-43 binds to *Nfasc* mRNA as seen from the TDP-43 mRNA binding CLIP-seq data from mouse brains, which shows binding peaks between exons 17 and 18 of *Nfasc* (upper panel). The exon color code in the upper panel follows that in **Figure 5A**; the bent arrows denote the two alternative transcription start sites from exons 1a and 1b (see **Supplementary file 1** for detail). The lower panel is a zoom into this region of interest, with tracks of *Figure 6 continued on next page*

*Figure 6 continued*

wild-type (WT) and TDP-43-cKO sciatic nerve RNA-seq read coverage and Sashimi plots to visualize major junction-spanning reads. The arc width is proportionate to the number of reads spanning the junction, which is labeled below each arc. There is a very large proportion of reads spanning from E17 to a new cryptic exonic region, upstream to the TDP-43-binding sites, in the cKO compared to the WT. (**B**) The RNA sequence transcribed from the cryptic exon (CE1) is given in purple. CE1 contains a stop codon in the open reading frame (ORF), well-defined splice acceptor and donor sites, and a downstream UG-rich TDP-43-binding region, characteristic of TDP-43 cryptic exons. (**C**) RT-PCR for *Nfasc* spliceforms using sciatic nerves and spinal cords isolated from P21 mice. Schematic of *Nfasc* spliceforms shows exon usage: E17–CE1–E19–E20 (i), E17–E18–E19–E20 (ii), and E17–E19–E20 (iii). The bands were excised and sequenced. Chromatograms covering the exon–exon junctions are shown at the left. Conditional heterozygote (cHet) and cKO by *Cnp-Cre*.

The online version of this article includes the following figure supplement(s) for figure 6:

**Figure supplement 1.** The identified cryptic exon is more highly expressed in sciatic nerves than in spinal cords and is spliced into >90% of *Nfasc* transcripts in the TDP-43-cKO sciatic nerves.

**Figure supplement 2.** TDP-43-cKO RNA-seq reads align to novel exonic region upstream of TDP-43-binding regions in *Nfasc*.

**Figure supplement 3.** The identified candidate cryptic exons contain a premature stop codon in the open reading frame (ORF).

paranodal junctions that are disrupted at 5 months post-tamoxifen injection is similar to the icKO recombination efficiency (*Figure 7D*), suggesting that paranodal disruption is prevalent among the mature Schwann cells lacking TDP-43. Altogether, our findings strongly suggest that TDP-43 in mature Schwann cells continues to repress the CE inclusion and supply NFasc for maintaining paranodal junctions.

## Discussion

TDP-43 is a ubiquitously expressed RNA/DNA-binding protein that covers one third of the transcriptome as potential targets of regulation (*Ederle and Dormann, 2017*; *Ling et al., 2013*). Consistent with its broad expression, emerging evidence indicates that the physiological and pathological roles for TDP-43 are no longer limited to neurons, but that TDP-43 is essential for normal function in glia (*Paolicelli et al., 2017*; *Peng et al., 2020*; *Velebit et al., 2020*; *Wang et al., 2018*). Additionally, the PNS has long been overlooked, though cell-autonomous roles for TDP-43 in peripheral cells have been implicated (*Gentile et al., 2019*). In this study, we demonstrate for the first time an essential role for TDP-43 in Schwann cells to facilitate rapid nerve conduction. We find that loss of TDP-43 in Schwann cells does not alter the structure of compact myelin, but instead the paranodal junction—the attachment site of myelin to the axon—is specifically disrupted, resulting in a 50% delay in nerve conduction velocity and functional motor deficits. Furthermore, we identify a previously unknown cryptic exon in the *Nfasc* gene, which overlaps with TDP-43-binding sites. Schwann cell NFasc protein is known to be essential for paranodal junction assembly and maintenance (*Pillai et al., 2009*; *Zonta et al., 2008*). We uncover the almost exclusive inclusion of this cryptic exon in the TDP-43-cKO Schwann cells, which introduces a premature stop codon and completely abolishes NFasc protein production, therefore, remarkably phenocopying NFasc cKO and icKO in Schwann cells (*Pillai et al., 2009*). The possession of a myriad of targets implies that TDP-43 is a pleiotropic gene, and we find that the myelinated axons in the TDP-43 cKO tend to have smaller diameters that may also contribute to the conduction delay (*Figure 1—figure supplement 2B*). However, from various studies, the absence of paranodal junctions in mice lacking Caspr or Schwann cell NFasc results in ~40–50% peripheral nerve conduction delay (*Bhat et al., 2001*; *Pillai et al., 2009*; *Susuki et al., 2013*), and we observe an average 52% delay in the TDP-43 cKO, suggesting that the absence of Schwann cell NFasc and paranodal junctions is the main contributor to the conduction delay in the TDP-43 cKO. All in all, our findings point to an absolute requirement of Schwann cell TDP-43 for NFasc expression and paranodal junction assembly/maintenance, and provide a framework for a mechanism where TDP-43 alters nerve conduction velocity by exerting its function in PNS myelin-forming glia to regulate neuron–glial interactions.

Schwann cell differentiation and myelination are controlled under a network of interacting transcription factors and signaling pathways (*Monk et al., 2015*; *Stolt and Wegner, 2016*). *Nfasc* and *Gldn* mRNA expression in Schwann cells coincides with the onset of myelination (*Basak et al., 2007*), suggesting that the transcriptional control for myelination also participates in orchestrating the formation of nodal and paranodal molecular domains during myelination. Nevertheless, the

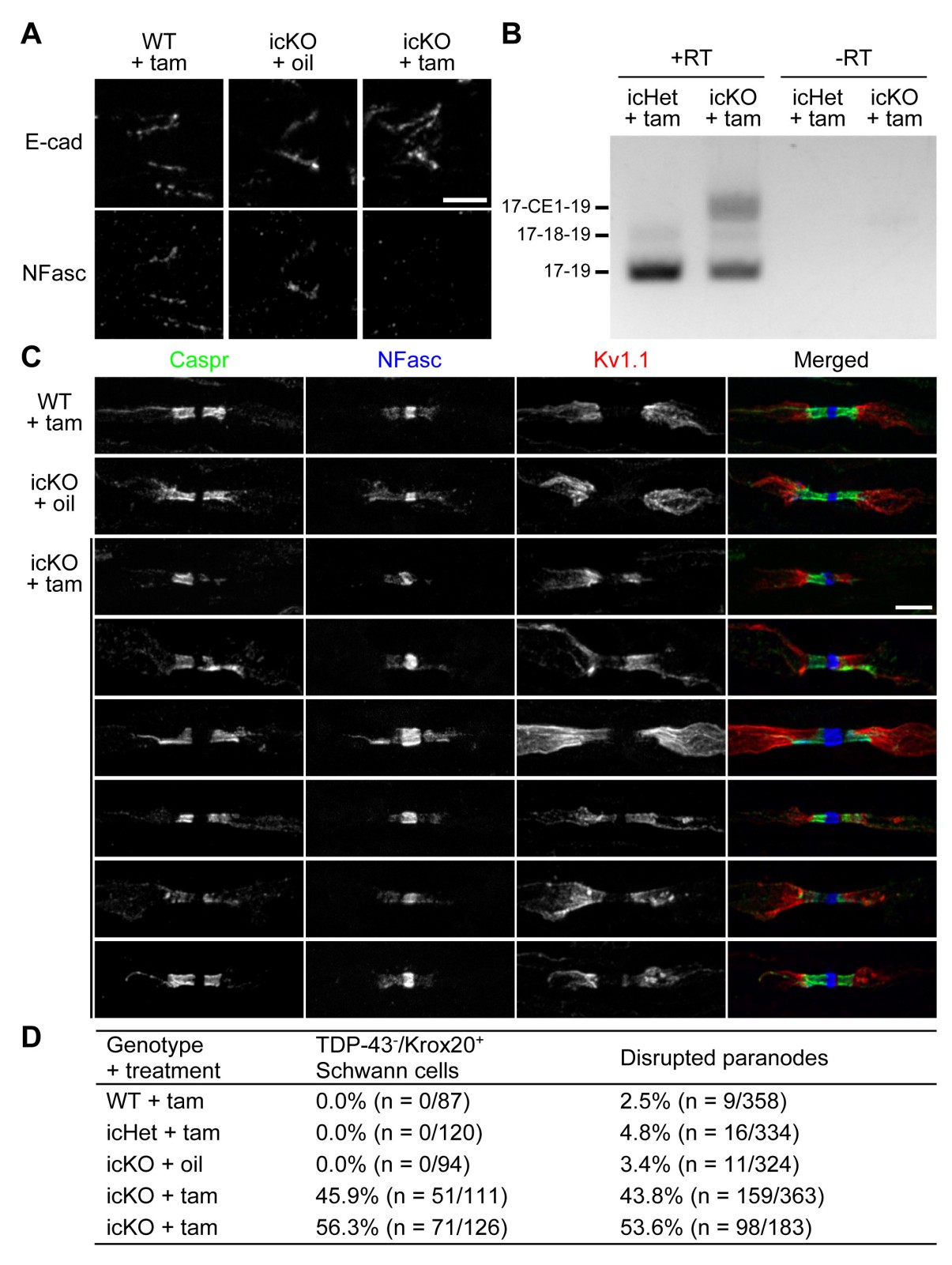

| Genotype + treatment | TDP-43⁻/Krox20⁺ Schwann cells | Disrupted paranodes |
|---|---|---|
| WT + tam | 0.0% (n = 0/87) | 2.5% (n = 9/358) |
| icHet + tam | 0.0% (n = 0/120) | 4.8% (n = 16/334) |
| icKO + oil | 0.0% (n = 0/94) | 3.4% (n = 11/324) |
| icKO + tam | 45.9% (n = 51/111) | 43.8% (n = 159/363) |
| icKO + tam | 56.3% (n = 71/126) | 53.6% (n = 98/183) |

**Figure 7.** Paranodal junctions are disrupted when TDP-43 is deleted in adult Schwann cells. (**A**) Immunostaining of adult sciatic nerves at 3 months after injection of tamoxifen (tam) or corn oil alone (oil) for E-cadherin (E-cad) (incisures) and neurofascin (NFasc). Scale bar, 5 μm. (**B**) RT-PCR for *Nfasc* spliceforms using sciatic nerves at 3 months after tamoxifen administration. +RT, reactions with reverse transcriptase; −RT, reactions without reverse transcriptase. (**C**) Immunostaining of adult sciatic nerves at 5 months after injection of tamoxifen (tam) or corn oil alone (oil) for contactin-associated

*Figure 7 continued on next page*

*Figure 7 continued*

protein (Caspr) (green), NFasc (blue), and Kv1.1 channel (red). Scale bar, 5 μm. (D) The sciatic nerve sections were immunostained for Krox20 and TDP-43 and quantified for the percentage of Krox20-positive nuclei that are TDP-43-negative. Immunostaining for Caspr and Kv1.1 was quantified for the percentage of Kv1.1-positive paranodes/juxtaparanodes that exhibit paranodal junction disruption defined as paranodal invasion of Kv1.1 and/or fragmented/shortened Caspr clusters. n represents the number of paranodes or nuclei analyzed for each mouse.

The online version of this article includes the following figure supplement(s) for figure 7:

**Figure supplement 1.** Neurofascin (NFasc) turns over faster at the incisures than at the paranodal junctions.

insufficiency of transcriptional regulation alone for controlling the precise timing of node assembly has recently been revealed for Gldn, whose node-inducing activity in premyelinating ensheathing Schwann cells is inhibited by Tolloid-like proteinases (*Eshed-Eisenbach et al., 2020*). Our identification of the novel cryptic exon residing in intron 17 of *Nfasc* may provide a fine-tuning mechanism governed by TDP-43, especially as the expression of TDP-43 is found to be highly regulated during development (*Sephton et al., 2010*) and may differ between cell types. Alternatively, TDP-43 may just play an all-or-none role in maintaining the intron integrity for *Nfasc* intron 17 to ensure NFasc expression. Future studies will determine whether and how the expression timing and paranodal stoichiometry of NFasc are regulated by TDP-43 and will provide further insight into the functional role of TDP-43 in the timely assembly and proper maintenance of paranodal junctions.

TDP-43 and NFasc are expressed by both Schwann cells and oligodendrocytes. Therefore, it is surprising to find that TDP-43 is required for NFasc expression and paranodal junction assembly exclusively in Schwann cells and not oligodendrocytes. Through our RNA-seq and splicing analyses, we also identify an increased usage of this *Nfasc* cryptic exon in the TDP-43-cKO spinal cords, although this usage is not increased to the same extent as in sciatic nerves. This suggests that TDP-43 actually performs in a similar manner on *Nfasc* expression by repressing this cryptic exon in both PNS and CNS myelinating glia. Conceivably, the difference in the extent of regulation in Schwann cells and oligodendrocytes may result from the different compositions of various RNA-binding proteins that regulate pre-mRNA splicing in different cell types (*Fu and Ares, 2014*). Future studies will identify the RNA-binding proteins in oligodendrocytes that may repress the *Nfasc* cryptic exon in the absence of TDP-43. Furthermore, neuronal NFasc (NF186) is highly enriched at the axon initial segments and nodes and is required for maintaining the high density of Nav channels therein (*Amor et al., 2017*; *Desmazieres et al., 2014*; *Zonta et al., 2011*). Ablation of neuronal NFasc drastically impairs the spontaneous firing of Purkinje neurons and delays peripheral nerve conduction. Given that this newly identified *Nfasc* cryptic exon is located immediately downstream of a constitutive exon shared by all NFasc isoforms, follow-up studies should investigate to what extent the expression level of neuronal NFasc is affected with the loss of function for TDP-43. Although not discussed in a previous study where TDP-43 is overexpressed under the Thy1.2 promoter to induce its aggregation in neurons, *Nfasc* is among the top-21 downregulated genes (*Shan et al., 2010*). Retrospectively, the direct regulation mechanism we uncover here may be generalizable to the physiology of neurons as well.

TDP-43 aggregates have been described in both neurons and glia in the CNS for over a decade (*Arai et al., 2006*; *Neumann et al., 2006*). It was not until recently that the pathological TDP-43 aggregates were discovered in Schwann cells in an ALS patient (*Nakamura-Shindo et al., 2020*). Our findings reveal that Schwann cell TDP-43 is essential for the formation and maintenance of paranodal junctions, and the cKO mice display motor deficits when challenged on the rotarod, suggesting that optimal nerve conduction in the PNS provided by Schwann cell TDP-43 and paranodal junctions is essential for motor coordination. Some distinct potential TDP-43-binding sites on human *NFASC* pre-mRNA could be identified using the human CLIP-seq data set (*Tollervey et al., 2011*) (data not shown). Therefore, follow-up studies are necessary to provide a comprehensive view on whether TDP-43 binds to *NFASC* transcripts and regulates *NFASC* expression in human Schwann cells, and to determine the prevalence of Schwann cell TDP-43 aggregates in patients and the functional consequence. Our findings are the first to demonstrate a functional role for TDP-43 in axon–glial interactions in the PNS and provide a framework and mechanism for how Schwann cell-autonomous dysfunction in nerve conduction is directly caused by TDP-43 loss of function.

# Materials and methods

## Key resources table

| Reagent type (species) or resource | Designation | Source or reference | Identifiers | Additional information |
|---|---|---|---|---|
| Genetic reagent (*Mus musculus*) | *Tardbp*<sup>fl/fl</sup> | Jackson Laboratory | Stock #: 017591; MGI:4834273 | |
| Genetic reagent (*M. musculus*) | *Dhh-Cre* | PMID:12782656 | MGI:4359600 | Dr. Dies Meijer |
| Genetic reagent (*M. musculus*) | *Mpz-Cre* (*P0-Cre*) | Jackson Laboratory | Stock #: 017927; MGI:2450448 | |
| Genetic reagent (*M. musculus*) | *Cnp-Cre* | PMID:12590258 | MGI:3051635 | Dr. Klaus-Armin Nave |
| Genetic reagent (*M. musculus*) | *Cntnap1*<sup>-/-</sup> (Caspr KO) | PMID:14676309 | MGI:3026869 | |
| Genetic reagent (*M. musculus*) | *Mpz-CreERT2* (*P0-CreERT2*) | PMID:12727441 | MGI:2663097 | Drs. Ueli Suter and Gabriel Corfas |
| Antibody | Anti-actin (mouse monoclonal C4) | MilliporeSigma | Cat #: MAB1501R; RRID:AB_2223041 | (1:5000) |
| Antibody | Anti-AnkB (mouse monoclonal N105/17) | UC Davis/NIH NeuroMab Facility | Clone: N105/17; RRID:AB_10674432 | (1:10) |
| Antibody | Anti-AnkG (mouse monoclonal N106/36) | UC Davis/NIH NeuroMab Facility | Clone: N106/36; RRID:AB_10697718 | (1:10) |
| Antibody | Anti-AnkG (mouse monoclonal N106/65) | UC Davis/NIH NeuroMab Facility | Clone: N106/65; RRID:AB_10673449 | (1:10) |
| Antibody | Anti-βII spectrin (mouse monoclonal 42) | BD Biosciences | Cat #: 612562; RRID:AB_399853 | (1:200) |
| Antibody | Anti-βIV spectrin (chicken polyclonal) | PMID:20980605 | RRID:AB_2827639 | (1:200) Dr. Matthew N. Rasband |
| Antibody | Anti-βIV spectrin (rabbit polyclonal) | PMID:15317849 | RRID:AB_2315634 | (1:500) Dr. Matthew N. Rasband |
| Antibody | Anti-Caspr (mouse monoclonal Mab275) | PMID:10624965 | RRID:AB_2314218 | (1:200) |
| Antibody | Anti-Caspr (rabbit polyclonal) | PMID:9118959 | RRID:AB_2314220 | (1:800 staining) |
| Antibody | Anti-Caspr (rabbit polyclonal) | Abcam | Cat#: ab34151; RRID:AB_869934 | (1:1000 blotting) |
| Antibody | Anti-Caspr2 (rabbit polyclonal) | PMID:10624965 | | (1:500) |
| Antibody | Anti-Cntn (goat polyclonal) | R&D Systems | Cat #: AF904-SP; RRID:AB_2292070 | (1:500) |
| Antibody | Anti-Cx32 (mouse monoclonal 5F9A9) | Thermo Fisher Scientific | Cat #: 35-8900; RRID:AB_2533228 | (1:200) |
| Antibody | Anti-E-cadherin (mouse monoclonal 36) | BD Biosciences | Cat #: 610181; RRID:AB_397580 | (1:200) |
| Antibody | Anti-Gldn (mouse monoclonal Mab94) | PMID:16039564 | | (1:200) |
| Antibody | Anti-Gldn (rabbit polyclonal) | PMID:17485493 | | (1:500) |
| Antibody | Anti-Krox20 (rabbit polyclonal) | PMID:15282162 | | (1:800) Dr. Dies Meijer |
| Antibody | Anti-Kv1.1 (mouse monoclonal K20/78) | UC Davis/NIH NeuroMab Facility | Clone: K20/78; RRID:AB_10672854 | (1:10) |

*Continued on next page*

Continued

| Reagent type (species) or resource | Designation | Source or reference | Identifiers | Additional information |
|---|---|---|---|---|
| Antibody | Anti-Kv1.2 (rabbit polyclonal) | PMID:7623158 | | (1:500) Dr. Matthew N. Rasband |
| Antibody | Anti-MAG (mouse monoclonal 513) | PMID:2444603 | | (1:500) Dr. Marie T. Filbin |
| Antibody | Anti-NFasc (chicken polyclonal) | R&D Systems | Cat #: AF3235; RRID:AB_10890736 | (1:200 staining) (1:500 blotting) |
| Antibody | Anti-panNav (mouse monoclonal K58/35) | MilliporeSigma | Cat #: S8809; RRID:AB_477552 | (1:200) |
| Antibody | Anti-PDGFRα (rabbit polyclonal) | PMID:8714519 | RRID:AB_2315173 | (1:2000) Dr. William B. Stallcup |
| Antibody | Anti-Sox10 (goat polyclonal) | R&D Systems | Cat #: AF2864; RRID:AB_442208 | (1:200) |
| Antibody | Anti-TDP-43 (mouse monoclonal 3H8) | EnCor Biotechnology | Cat #: MCA-3H8; RRID:AB_2572387 | (1:800) |
| Antibody | Anti-TDP-43 (rabbit polyclonal) | Proteintech | Cat #: 10782-2-AP; RRID:AB_615042 | (1:500) |
| Antibody | Anti-ZO-1 (mouse monoclonal 1A12) | Thermo Fisher Scientific | Cat #: 33-9100; RRID:AB_2533147 | (1:200) |
| Antibody | Anti-chicken AMCA (goat polyclonal) | Jackson ImmunoResearch | Cat#: 103-155-155; RRID:AB_2337385 | (1:200) |
| Antibody | Anti-chicken AlexaFluor 594 (goat polyclonal) | Thermo Fisher Scientific | Cat#: A-11042; RRID:AB_2534099 | (1:1000) |
| Antibody | Anti-chicken DyLight 680 (goat polyclonal) | Rockland | Cat #: 603-144-126; RRID:AB_1057473 | (1:10,000) |
| Antibody | Anti-goat AlexaFluor 594 (donkey polyclonal) | Thermo Fisher Scientific | Cat#: A-11058; RRID:AB_2534105 | (1:1000) |
| Antibody | Anti-mouse IRDye 800CW (goat polyclonal) | LI-COR Biotechnology | Cat #: 925-32210; RRID:AB_2687825 | (1:10,000) |
| Antibody | Anti-mouse IgG1 AlexaFluor 488 (goat polyclonal) | Thermo Fisher Scientific | Cat#: A-21121; RRID:AB_2535764 | (1:1000) |
| Antibody | Anti-mouse IgG1 AlexaFluor 594 (goat polyclonal) | Thermo Fisher Scientific | Cat#: A-21125; RRID:AB_2535767 | (1:1000) |
| Antibody | Anti-mouse IgG1 AlexaFluor 647 (goat polyclonal) | Thermo Fisher Scientific | Cat#: A-21240; RRID:AB_2535809 | (1:1000) |
| Antibody | Anti-mouse IgG2a AlexaFluor 488 (goat polyclonal) | Thermo Fisher Scientific | Cat#: A-21131; RRID:AB_2535771 | (1:1000) |
| Antibody | Anti-mouse IgG2a AlexaFluor 594 (goat polyclonal) | Jackson ImmunoResearch | Cat#: 115-585-206; RRID:AB_2338886 | (1:800) |
| Antibody | anti-mouse IgG2a AlexaFluor 647 (goat polyclonal) | Thermo Fisher Scientific | Cat#: A-21241; RRID:AB_2535810 | (1:1000) |
| Antibody | Anti-mouse IgG2b AlexaFluor 594 (goat polyclonal) | Thermo Fisher Scientific | Cat#: A-21145; RRID:AB_2535781 | (1:1000) |

*Continued*

| Reagent type (species) or resource | Designation | Source or reference | Identifiers | Additional information |
|---|---|---|---|---|
| Antibody | Anti-mouse IgG2b AlexaFluor 647 (goat polyclonal) | Thermo Fisher Scientific | Cat#: A-21242; RRID:AB_2535811 | (1:1000) |
| Antibody | Anti-rabbit AlexaFluor 488 (donkey polyclonal) | Thermo Fisher Scientific | Cat#: A-21206; RRID:AB_2535792 | (1:1000) |
| Antibody | Anti-rabbit AlexaFluor 488 (goat polyclonal) | Thermo Fisher Scientific | Cat#: A-11034; RRID:AB_2576217 | (1:1000) |
| Antibody | Anti-rabbit AlexaFluor 594 (goat polyclonal) | Thermo Fisher Scientific | Cat#: A-11037; RRID:AB_253409 | (1:1000) |
| Antibody | Anti-rabbit IRDye 800CW (goat polyclonal) | LI-COR Biotechnology | Cat #: 925-32211; RRID:AB_2651127 | (1:10,000) |

## Mice

Construction of the mouse lines was described previously: *Tardbp$^{fl/fl}$* (*Chiang et al., 2010*) (JAX 017591); *Dhh-Cre* (*Jaegle et al., 2003*); *Mpz-Cre* (*P0-Cre*) (*Feltri et al., 1999*) (JAX 017927); *Cnp-Cre* (*Lappe-Siefke et al., 2003*); *Cntnap1$^{-/-}$* (Caspr KO) (*Gollan et al., 2003*); *Mpz-CreERT2* (*P0-CreERT2*) (*Leone et al., 2003*). All experiments involving mice were performed in compliance with the National Institutes of Health Guide for the Care and Use of Laboratory Animals or the Association for Assessment and Accreditation of Laboratory Animal Care guidelines for animal use and were preapproved by the Institutional Animal Care and Use Committees at the University of California, San Francisco (protocol number: AN180003), National University of Singapore (protocol numbers: BR17-0928 and R17-0634), and Wright State University (Animal Use Protocol # 1113). The mice were housed in barrier facilities under a 12 hr light/dark cycle with free access to food and water. Male and female mice were randomly allocated to the experimental groups according to their age and genotypes. For the icKO experiments, tamoxifen was dissolved in corn oil at 20 mg/ml by shaking at 37°C overnight, and 0.1–0.14 ml (100 mg tamoxifen/kg body weight) was administered intraperitoneally once per day for five consecutive days.

## Antibodies

The following primary antibodies were used: mouse anti-actin (MilliporeSigma, C4), mouse anti-AnkB (UC Davis/NIH NeuroMab Facility, N105/17), mouse anti-AnkG (NeuroMab, N106/36), mouse anti-AnkG (NeuroMab, N106/65), mouse anti-βII spectrin (BD Biosciences, 42), chicken anti-βIV spectrin (a gift from Dr. Matthew N. Rasband), rabbit anti-βIV spectrin (a gift from Dr. Matthew N. Rasband, Baylor College of Medicine), mouse anti-Caspr (Mab275) (*Poliak et al., 1999*), rabbit anti-Caspr (*Peles et al., 1997*), rabbit anti-Caspr (Abcam ab34151), rabbit anti-Caspr2 (*Poliak et al., 1999*), goat anti-Cntn (R&D Systems, AF904), mouse anti-Cx32 (Thermo Fisher Scientific, 5F9A9), mouse anti-E-cadherin (BD Biosciences, 36), mouse anti-Gldn (Mab94) (*Eshed et al., 2005*), rabbit anti-Gldn (*Eshed et al., 2007*), rabbit anti-Krox20 (a gift from Dr. Dies Meijer, University of Edinburgh), mouse anti-Kv1.1 channel (NeuroMab, K20/78), rabbit anti-Kv1.2 channel (a gift from Dr. Matthew N. Rasband), mouse anti-MAG (513) (a gift from Dr. Marie T. Filbin, The City University of New York), chicken anti-NFasc (R&D Systems, AF3235), mouse anti-panNav channels (MilliporeSigma, K58/35), rabbit anti-PDGFRα (a gift from Dr. William B. Stallcup, Sanford Burnham Prebys Medical Discovery Institute), goat anti-Sox10 (R&D Systems, AF2864), mouse anti-TDP-43 (EnCor Biotechnology, 3H8), rabbit anti-TDP-43 (Proteintech, 10782-2-AP), and mouse anti-ZO-1 (Thermo Fisher Scientific, 1A12). The secondary antibodies were purchased from Thermo Fisher Scientific, LI-COR Biotechnology, and Jackson ImmunoResearch Laboratories.

## Immunostaining

The sciatic nerves, trigeminal nerves, and spinal cords were either dissected after transcardial perfusion and postfixed as described previously (*Wang et al., 2018*) or freshly dissected and then fixed in

4% (w/v) paraformaldehyde (PFA) in 0.1 M sodium phosphate (PB), pH7.4, for 30 min (nerves) or 1 hr (spinal cords) on ice. Fixed tissues were immersed in 20% (w/v) sucrose in 0.1 M PB at 4°C overnight and embedded in a mixture containing 5% (w/v) sucrose, 25 mM PB, and 75% (v/v) O.C.T. Compound (Sakura Finetek) for cryosectioning. Tissue sections were spread onto gelatin-coated coverslips. Alternatively, fixed nerves or spinal nerve roots were immersed in 150 mM sodium chloride, 10 mM sodium phosphate (PBS), pH 7.2, and teased on gelatin-coated coverslips. The teased nerve fibers were permeabilized in 100% methanol at −20°C for 10 min and rinsed with PBS three times before subjected to the subsequent staining procedure.

After air dried, the tissues on the coverslips were blocked with 10% (v/v) normal goat or donkey serum, 0.3% (v/v) Triton X-100 in PBS for 30 min, and stained with primary antibodies diluted in the blocking solution at 4°C overnight. After one wash with the blocking solution and two washes with PBS for 5 min each, the tissues were stained with secondary antibodies diluted in the blocking solution at room temperature for 1 hr. After one wash with the blocking solution and two washes with PBS, the coverslips were mounted onto glass slides in Vectashield Vibrance mountant (Vector Laboratories) or ProLong Diamond mountant (Thermo Fisher Scientific).

## Electrophysiology

Nerve conduction studies were performed as described previously with modification (*Otani et al., 2017*). In brief, the sciatic nerve and its tibial branch were stimulated by needle electrodes inserted close to the nerve at the ankle and sciatic notch under general anesthesia with 2% (v/v) isoflurane inhalation. Supramaximal stimulations were used, and the evoked compound muscle action potentials were recorded from the plantar muscles through needle electrodes placed transversely over the muscle bellies in the sole of the foot. Motor nerve conduction velocity was calculated by dividing the distance between the ankle and sciatic notch by the difference in latency between the ankle and sciatic notch.

## Transmission electron microscopy

After sacrificing the mice, the sciatic nerves were exposed and the lower bodies of the mice were fixed in 4% (w/v) PFA, 2.5% (w/v) glutaraldehyde in 0.1 M sodium cacodylate, 5 mM calcium chloride, pH 7.4 at room temperature overnight, and at 4°C for 2 days. Nerves were dissected and stored in the same fixative at 4°C for at least 2 days. Nerves were then processed as previously described (*Feinberg et al., 2010*) and were examined using an FEI Tecnai T12 transmission electron microscope or Tecnai F20 S/TEM equipped with an XF416 TVIP camera or a US4000 Gatan camera, respectively.

## Rotarod analysis

Four-week-old mice were first familiarized with the rotarod (Ugo Basile 47600 V04) at 4 rpm for 5 min. Thirty minutes later, the mice were challenged with accelerating rotarod from 4 to 40 rpm within 5 min for three trials, separated by 30 min. The next day, the mice were tested with the rotation speed increased from 4 rpm to 24 or 32 rpm within 30 s and maintained at 24 or 32 rpm, respectively, for another 60 s. Three trials were performed for each speed, and the mice were allowed to rest for at least 10 min between trials. The time periods for which each mouse remained on the rotarod at each speed were recorded and averaged.

## Preparation of nerve homogenates for western blotting

Sciatic nerves were homogenized with sonication in 320 mM sucrose, 5 mM sodium phosphate (pH 7.2), 0.2 mM sodium fluoride, 0.2 mM sodium orthovanadate, 1× cOmplete protease inhibitor cocktail (Roche), and 1 mM phenylmethylsulfonyl fluoride on ice. The protein concentration was determined with the BCA protein assay kit (Thermo Fisher Scientific) using the standard curve derived from a serial dilution of bovine serum albumin; 30 µg of total protein was subjected to western blotting.

## Reverse transcription-quantitative polymerase chain reaction (RT-qPCR)

Sciatic nerves were homogenized in the TRIzol reagent (Thermo Fisher Scientific) using a Dounce homogenizer (Wheaton) and incubated at room temperature for 5 min. The homogenate was

extracted with chloroform, incubated at room temperature for 3 min, and centrifuged at 12,000×$g$ for 15 min at 4℃. The total RNA was purified from the supernatant using the Direct-zol RNA Miniprep kit (Zymo Research) by following the manufacturer's instruction including the in-column DNase I treatment. 0.5 µg of total RNA (concentration determined by the absorbance at 260 nm) was used in 20 µl of reverse transcription reaction with LunaScript RT SuperMix (New England Biolabs) and its No-RT Control Mix according to the manufacturer's instruction. The RT and No-RT reactions were then diluted to 1/4× with deionized water, and 2 µl was used for qPCR in 25 µl of reaction containing 0.5 µM primers with Power SYBR Green PCR Master Mix and QuantStudio 3 Real-Time PCR System (Thermo Fisher Scientific). The thermocycling parameters are 95℃ for 10 min, and 40 cycles of 95℃ for 15 s and 60℃ for 1 min, followed by melting curve analysis. The primers used in qPCR are as follows (from 5′ to 3′): *Tardbp* ex3 (forward: GGAGAGGTTCTTATGGTTCAGG; reverse: TTCTCAAAGGCTCGTCTGG; 187 bp), *Tardbp* ex4-5 (forward: TGGTGTGACTGTAAACTTCCC; reverse: CGAAGGCAAAAGCTCTGAATG; 181 bp), pan *Nfasc* (forward: CAGTGGATGGTGAATGGAG; reverse: GCAGGTAGCCATGTTCATTG; 148 bp), NF155 (forward: CCTGTCTACGTTCCCTATGAG; reverse: GTTCCACTGAAGGCTGATG; 168 bp), NF140+ex27 (forward: GGAGAAGATTTACCCAGTGC; reverse: TATTGGTGTAAGCTGCAGTT; 334 bp), NF140-ex27 (forward: the same as NF140+ex27; reverse: GGTTATTGGTGTAAGCTTCATTT; 322 bp), *G6pdx* (forward: CCTCAACAGCCACATGAATG; reverse: GGTTCGACAGTTGATTGGAG; 197 bp), and *Polr2a* (forward: CATCAAGAGAGTGCAGTTCG; reverse: CCATTAGTCCCCCAAGTTTG; 125 bp). *Polr2a* was chosen as the reference gene (*Radonić et al., 2004*), and the standard deviations of the *Polr2a* $C_q$ values among WT, cHet, and cKO ranged from 0.100 to 0.131. We also tested the expression consistency of *G6pdx* and found that *Polr2a* outcompetes *G6pdx* (the *G6pdx* $C_q$ standard deviations among WT, cHet ,and cKO were up to 0.183). The efficiency of each primer pair was validated to be between 90% and 110% with six points of threefold or fourfold serial dilutions of P16 mouse sciatic nerve cDNAs or P31 mouse spinal cord cDNAs. The primer specificity was screened on the Ensembl genome browser by using the BLASTN search tool against the cDNA database and genomic sequence, and confirmed by gel electrophoresis and sequencing from both ends of the PCR products. Technical triplicates of the RT reaction and duplicates of the No-RT reaction were performed for each biological replicate, and three biological replicates were analyzed. The relative transcript level was calculated by the ΔΔ$C_q$ method. Comparison of cHet or cKO with WT was carried out by one-sample unpaired two-tailed *t*-tests with the test mean equal to 1.

## RT-PCR

Total RNA was extracted from sciatic nerves and spinal cords using the TRIzol reagent according to manufacturer's instruction. After DNase treatment using RQ1 RNase-Free DNase (Promega), 0.5–1 µg of RNA was reverse transcribed using Maxima First Strand cDNA Synthesis Kit for RT-qPCR (Thermo Fisher Scientific) and subsequently used for PCR. Primers for mouse *Nfasc* gene spanning exons 17 and 20 were (from 5′ to 3′): GCAAAGGCCTACCTCACTGT (forward) and CTCGTTGACAGCGATGACTC (reverse). Similar results were also obtained by using GACCAGGGCAGTTACACG (forward, exon 17) and TCACACTCCTCTCAGCCAG (reverse, exon 19). The PCR products were resolved in agarose gel.

## RNA-seq and cryptic exon analysis

RNA extracted from the sciatic nerves of 21-day-old *Tardbp*<sup>fl/fl</sup>, *Cnp-Cre;Tardbp*<sup>fl/+</sup>, and *Cnp-Cre; Tardbp*<sup>fl/fl</sup> mice (n = 3 per genotype) were used to prepare RNA-seq libraries using the TruSeq RNA Sample Prep Kit (Illumina). Illumina HiSeq4000 was used for paired-end 151 bp sequencing and yielded 30–40 million reads per sample. This data has been deposited in NCBI's Gene Expression Omnibus (*Edgar et al., 2002*) and is accessible through GEO Series GSE157714.

To identify novel cryptic exons present in the data, genomic alignment BAM files were generated using HISAT2 v.2.1.0 (*Kim et al., 2015*), aligned to mouse reference genome GRCm38 (Ensembl 94 release) (*Cunningham et al., 2019*). The RNA-seq reads aligned to *Nfasc* were visualized using the Bioconductor R package Gviz v.1.28.3 (*Hahne and Ivanek, 2016*). TDP-43 CLIP-seq data from adult mouse brains from Yeo and Cleveland groups (GEO Series GSE40651) (*Polymenidou et al., 2011*) was used to visualize TDP-43 mRNA binding sites. The number of reads spanning different exon–exon junctions was extracted from the alignment BAM files using Regtools v.0.5.1 junctions extract

function (*Feng et al., 2018*). This information was then used to calculate the PSI values for junctions of interest. PSI values reflect the efficiency of a specific exon being spliced in the transcript population and are calculated as the ratio of inclusion reads (reads spliced to exon of interest) to the sum of inclusion and exclusion reads (reads spliced to other exons).

## Statistical analysis

Except electrophysiological recordings and electron microscopy, the other data analysis was not performed in a blinded way. Sample size was not determined in advance using any statistical method, but is comparable to the one used in the literature of the field (*Susuki et al., 2018*). Sets of age-matched WT, cHet, and cKO mice from the same litter or from two litters that had the closest dates of birth were randomly assigned together for analysis. The gender of the mice was recorded and not controlled; the gender did not affect the reproducibility of our study. One-way analysis of variance (ANOVA) tests with Tukey's post-hoc analysis, Mann–Whitney tests, and unpaired two-tailed *t*-tests were conducted using OriginPro (OriginLab). The data points of five or less were assumed to be drawn from normal distributions. For the ANOVA tests, homogeneity of variance was not rejected by Levene's tests. Biological replication is achieved by analyzing different mice. Technical replication is achieved by repeating the experiments using the same samples. The number of mice or technical replicates used for each comparison is described in the figure legends and Materials and methods. No data points were excluded except that one WT mouse in the rotarod test was considered an outlier, which may have learned that the consequences of falling are innocuous (*Brooks and Dunnett, 2009*). Using the interquartile range method, it is far below the lower outlier fence ( = Q1 − IQR ×1.5). The detailed information on this excluded mouse can be found in *Figure 2—source data 1*.

## Acknowledgements

We thank Dr. Ueli Suter (ETH Zurich) for providing the *Mpz-CreERT2* mouse line, Dr. Dies Meijer (University of Edinburgh) for the *Dhh-Cre* mouse line, Dr. Klaus-Armin Nave (Max Planck Institute of Experimental Medicine) for the *Cnp-Cre* mouse line, and Dr. Scott S Zamvil (UCSF) for the access to the rotarod apparatus. We thank all the members of the Chan laboratory for many insightful discussions, advice, and encouragement. This work was supported by funding from the National Institutes of Health/National Institute of Neurological Disorders and Stroke (grants R01NS062796, R01NS097428, and R01NS095889), the Dr. Miriam and Sheldon G Adelson Medical Research Foundation (APND grant A130141), the Rachleff family endowment to JRC, the Israel Science Foundation and the Dr. Miriam and Sheldon G Adelson Medical Research Foundation to EP, the National Medical Research Council, Singapore (NMRC/OFIRG/0001/2016 and NMRC/OFIRG/0042/2017), and the Swee Liew-Wadsworth Endowment fund, National University of Singapore to SCL, the National Institutes of Health/National Institute of Neurological Disorders and Stroke (grants R01NS107398 and R03NS112981) to KS, and the National Multiple Sclerosis Society Postdoctoral Fellowship (FG-1507-05195) to KJC. EP is the Incumbent of the Hanna Hertz Professorial Chair for Multiple Sclerosis and Neuroscience.

## Additional information

### Funding

| Funder | Grant reference number | Author |
| --- | --- | --- |
| National Institute of Neurological Disorders and Stroke | R01NS062796 | Jonah R Chan |
| National Institute of Neurological Disorders and Stroke | R01NS097428 | Jonah R Chan |
| National Institute of Neurological Disorders and Stroke | R01NS095889 | Jonah R Chan |
| Dr. Miriam and Sheldon G. Adelson Medical Research Foundation | APND grant A130141 | Jonah R Chan |

| | | |
|---|---|---|
| Rachleff Family Endowment | | Jonah R Chan |
| National Medical Research Council | NMRC/OFIRG/0001/2016 | Shuo-Chien Ling |
| National Medical Research Council | NMRC/OFIRG/0042/2017 | Shuo-Chien Ling |
| Swee Liew-Wadsworth Endowment fund | | Shuo-Chien Ling |
| Israel Science Foundation | | Elior Peles |
| Dr. Miriam and Sheldon G. Adelson Medical Research Foundation | | Elior Peles |
| National Institute of Neurological Disorders and Stroke | R01NS107398 | Keiichiro Susuki |
| National Institute of Neurological Disorders and Stroke | R03NS112981 | Keiichiro Susuki |
| National Multiple Sclerosis Society | FG-1507-05195 | Kae-Jiun Chang |

The funders had no role in study design, data collection and interpretation, or the decision to submit the work for publication.

## Author contributions

Kae-Jiun Chang, Conceptualization, Data curation, Formal analysis, Funding acquisition, Validation, Investigation, Visualization, Methodology, Writing - original draft, Writing - review and editing; Ira Agrawal, Conceptualization, Data curation, Formal analysis, Validation, Investigation, Methodology, Writing - original draft; Anna Vainshtein, Data curation, Formal analysis, Investigation; Wan Yun Ho, Data curation, Formal analysis, Investigation, Methodology; Wendy Xin, Data curation, Formal analysis, Investigation, Writing - review and editing; Greg Tucker-Kellogg, Formal analysis, Supervision, Investigation, Methodology, Writing - original draft; Keiichiro Susuki, Data curation, Formal analysis, Investigation, Methodology, Writing - review and editing; Elior Peles, Formal analysis, Supervision, Project administration, Writing - review and editing; Shuo-Chien Ling, Conceptualization, Formal analysis, Supervision, Funding acquisition, Investigation, Project administration, Writing - review and editing; Jonah R Chan, Conceptualization, Supervision, Funding acquisition, Investigation, Project administration, Writing - review and editing

## Author ORCIDs

Kae-Jiun Chang https://orcid.org/0000-0002-3656-7725
Ira Agrawal http://orcid.org/0000-0002-2195-6090
Wendy Xin http://orcid.org/0000-0002-1425-834X
Shuo-Chien Ling https://orcid.org/0000-0002-0300-8812
Jonah R Chan https://orcid.org/0000-0002-2176-1242

## Ethics

Animal experimentation: All experiments involving mice were performed in compliance with the National Institutes of Health Guide for the Care and Use of Laboratory Animals or the Association for Assessment and Accreditation of Laboratory Animal Care guidelines for animal use, and were pre-approved by the Institutional Animal Care and Use Committees at the University of California, San Francisco (protocol number: AN180003), National University of Singapore (protocol numbers: BR17-0928 and R17-0634), and Wright State University (Animal Use Protocol # 1113).

## Decision letter and Author response

Decision letter https://doi.org/10.7554/eLife.64456.sa1
Author response https://doi.org/10.7554/eLife.64456.sa2

# Additional files

## Supplementary files

• Supplementary file 1. The exon nomenclature of *Nfasc* used in this study. The exon information was compiled from the current Ensembl database. The coordinates of each exon in chromosome 1 are shown (GRCm38.p6). Transcription starts from either exon 1a or 1b and terminates at either exon 23T or 33. The alternative exons specific for NF155 (exons 23 or 23L and 24) and the ones for NF186 (exons 28–30) are shaded in blue. Other alternative exons are shaded in gray. The cryptic exon CE1 identified in this study is shaded in pink. The start codon and potential stop codons are framed in red.

• Transparent reporting form

## Data availability

NCBI Gene Expression Omnibus, GSE157714.

The following previously published dataset was used:

| Author(s) | Year | Dataset title | Dataset URL | Database and Identifier |
|---|---|---|---|---|
| Yeo GW, Cleveland DW | 2011 | Divergent roles of ALS-linked proteins FUS/TLS and TDP-43 intersect in processing long pre-mRNAs (CLIP-Seq) | https://www.ncbi.nlm.nih.gov/geo/query/acc.cgi?acc=GSE40651 | NCBI Gene Expression Omnibus, GSE40651 |

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
