## [Decision Letter]

**Acceptance summary:**

During development, selective deletion of TDP-43 from Schwann cells revealed a cell-autonomous function in paranodal junction assembly and motor nerve conduction, but not formation of compact myelin. Mechanistic studies identified that the production of the glia-specific splice form of *Nfasc* (encoding NF155) is reduced, while the neuronal splice form encoding NF186 is unaltered. While it is likely that TDP-43 affects additional transcripts in Schwann cells, this study reveals one key protein that is altered as a consequence of TDP-43 deletion. As loss of TDP-43 function is associated with numerous disease and injury states, the studies have important implications for the consequences of these changes in Schwann cells.

**Decision letter after peer review:**

Thank you for submitting your article "TDP-43 maximizes nerve conduction velocity by repressing a cryptic exon for paranodal junction assembly in Schwann cells" for consideration by *eLife*. Your article has been reviewed by three peer reviewers, and the evaluation has been overseen by a Reviewing Editor and Gary Westbrook as the Senior Editor. The following individuals involved in review of your submission have agreed to reveal their identity: Roman Giger (Reviewer #1); James Salzer (Reviewer #2); Phil Wong (Reviewer #3). The reviewers have discussed the reviews with one another and the Reviewing Editor has drafted this decision to help you prepare a revised submission.

Summary:

This study describes the consequences of selective deletion of the of the RNA binding protein TDP-43 from Schwann cells. Using in vivo genetic manipulations in mice, the authors show that TDP-43 is required for both the formation and maintenance of paranodal junctions in the peripheral nervous system. TDP-43 is known to suppress the incorporation of cryptic exons in diverse mRNAs. In Schwann cells lacking TDP-43 they find that inclusion of a cryptic exon in *Nfasc* mRNA (which encodes the paranodal protein NF155) results in a premature stop codon, triggering nonsense-mediated decay of its mRNA. The absence of this protein provides an explanation for the altered assembly of paranodal junctions, the marked delay in peripheral nerve conduction and associated motor discoordination. Nuclear exclusion of TDP-43 occurs in many cell types following injury and progressive disease, raising the possibility that Schwann cell dysfunction may also contribute to pathological changes during these conditions.

Essential revisions:

1) Provide information about the characteristics of axons and myelin in the Tardbp cko mice (g-ratio, axon diameters, etc). Changes in these features could also affect conduction velocities and contribute to paranodal defects. This is particularly important, given the large number of potential mRNA targets of TDP-43 and the observation that myelination is delayed in the conditional mutants, a phenomenon not normally observed in paranodal mutants. If this analysis is not possible in a reasonable time frame, please describe in detail the observations that can be made from existing electron micrographs and revise the text to discuss how alterations in other aspects of these myelinated nerves could contribute to the phenotypes exhibited by the Tardbp cKO mice.

2) Revise Figure 1. It is difficult to appreciate that TDP-43 is present in normal Schwann cells without displaying the Sox10 channel alone. Quantification of overlap should be provided, along with evidence that nuclear TDP-43 in non-Schwann cells was not affected by this manipulation. In Figure 1A it appears that Sox10 is strongly upregulated in cKO nerves. If this is the case, provide quantification, if not, replace figure panel with a more representative example (or single channel images). Dhh-Cre is active in endoneurial fibroblast precursors (Joseph et al., 2004), so it is unclear what cell type persistently express TDP43 in the P28 Dhh-Cre cKOs. The quality of the images of isolated nerves in panel D could be improved.

3) Revise the discussion of TDP-43 splicing repression in human Schwann cells. The lack of conservation between cryptic exons between species make this hypothesis very speculative. It would enhance the impact of the study to search available databases for evidence in humans of TDP-43 targets related to paranodal junction genes. Such human data would provide strong clinical implications of findings from this manuscript. It is possible that other TDP-43 targets related to paranodal junction are impacted in ALS patients. However, inclusion of these data is not deemed necessary for publication.

4) Revise Figure 2. Figure 2H: In assessing behavioral outcomes of TDP-43 cKO mice, a mild defect in rotarod test was observed and it appeared that there is a bimodal distribution where one group of mice show a deficit whereas the other group seemed normal. Can this be explained by extent of neurofascin cryptic exon inclusion or gender of mice? Is there assessment using an independent behavioral task to confirm this finding? Given the non-normal distribution, a standard t-test is not appropriate for this statistical comparison.

---

## [Author Response]

Essential revisions:1) Provide information about the characteristics of axons and myelin in the Tardbp cko mice (g-ratio, axon diameters, etc). Changes in these features could also affect conduction velocities and contribute to paranodal defects. This is particularly important, given the large number of potential mRNA targets of TDP-43 and the observation that myelination is delayed in the conditional mutants, a phenomenon not normally observed in paranodal mutants. If this analysis is not possible in a reasonable time frame, please describe in detail the observations that can be made from existing electron micrographs and revise the text to discuss how alterations in other aspects of these myelinated nerves could contribute to the phenotypes exhibited by the Tardbp cKO mice.

We want to thank the reviewer for this important comment, as it constitutes an alternative conclusion. We have now measured the g-ratios and axon diameters from sciatic nerves of three WT and three cKO mice at P21, and the data is now presented in Figure 1—figure supplement 2. The g-ratios between the WT and cKO mice are not statistically different (p = 0.1214 by *t* test). The myelin thickness of the WT and cKO is comparable and does not contribute to the conduction delay along the cKO nerves. We find that the myelinated axons in the cKO tend to have smaller diameters; however, because the absence of paranodal junctions is known to cause ~40-50% conduction delay (≥ 50% delay in Caspr KO [Susuki et al., 2013], 41-42% delay in Caspr KO [Bhat et al., 2001], 41% delay in the *Nfasc* cKO by P0-Cre [Pillai et al., 2009]), and we observed a 52% delay in the TDP-43 cKO, we believe that the smaller axon diameter in the cKO only contributes to a small portion in the delay of nerve conduction. We have revised the manuscript to discuss this variable.

2) Revise Figure 1. It is difficult to appreciate that TDP-43 is present in normal Schwann cells without displaying the Sox10 channel alone. Quantification of overlap should be provided, along with evidence that nuclear TDP-43 in non-Schwann cells was not affected by this manipulation. In Figure 1A it appears that Sox10 is strongly upregulated in cKO nerves. If this is the case, provide quantification, if not, replace figure panel with a more representative example (or single channel images). Dhh-Cre is active in endoneurial fibroblast precursors (Joseph et al., 2004), so it is unclear what cell type persistently express TDP43 in the P28 Dhh-Cre cKOs. The quality of the images of isolated nerves in panel D could be improved.

We now include the single-channel images for TDP-43 and Sox10 from P28 in Figure 1—figure supplement 1, together with those from P3. All the Sox10-positive nuclei are TDP-43-positive in the WT. The intensity of Sox10 staining varies among different nuclei of the WT and cKO especially at P28, and we did not observe a change in the Sox10 levels in the cKO.

We thank the reviewers for bringing our attention to the study by Joseph et al., 2004. We observed the complete loss of paranodal junctions in the PNS using three different Cre driver lines to conditionally knock out TDP-43 in our study—Dhh-Cre, P0-Cre and Cnp-Cre—and this strongly indicates that the phenotype we observe is cell-autonomous to Schwann cells. Furthermore, we have stained the endoneurial fibroblasts with PDGFRα and confirmed that the fibroblasts are among the TDP-43-positive/Sox10-negative population in the cKO mice (by Dhh-Cre), which includes other PDGFRα-negative, non-fibroblast cells as well (Figure 1—figure supplement 1C). The apparent discrepancy between our observation and that of Joseph et al. with the Dhh-Cre in endoneurial fibroblasts may be attributed to the different floxed alleles involved. This has been shown to be the case for Cnp-Cre (Tognatta et al., 2017 [PMID: 27807896] showed that the recombination events by Cnp-Cre in neurons or astrocytes are restricted to specific reporter alleles). In brief, when combining the Tardbp floxed allele and Dhh-Cre, we do not observe recombination in endoneurial fibroblasts.

We have changed panel D of Figure 1 to improve the picture quality.

3) Revise the discussion of TDP-43 splicing repression in human Schwann cells. The lack of conservation between cryptic exons between species make this hypothesis very speculative. It would enhance the impact of the study to search available databases for evidence in humans of TDP-43 targets related to paranodal junction genes. Such human data would provide strong clinical implications of findings from this manuscript. It is possible that other TDP-43 targets related to paranodal junction are impacted in ALS patients. However, inclusion of these data is not deemed necessary for publication.

We have analyzed the human TDP-43 CLIP-seq dataset by Tollervey et al., 2011, and found several potential TDP-43-binding sites on human *NFASC* pre-mRNA (see Author response image 1). At first glance, the binding sites for TDP-43 in humans are distinct from mice, but the human dataset has a much lower coverage than the mouse; therefore, further experiments are needed to provide a comprehensive view on whether TDP-43 binds to *NFASC* transcripts and regulates *NFASC* expression in human Schwann cells. We believe that this is a really important point but is beyond the scope of our current study. We anticipate that future investigation with human Schwann cells will be exceedingly interesting and will help determine the relevance of our study to the human condition.

4) Revise Figure 2. Figure 2H: In assessing behavioral outcomes of TDP-43 cKO mice, a mild defect in rotarod test was observed and it appeared that there is a bimodal distribution where one group of mice show a deficit whereas the other group seemed normal. Can this be explained by extent of neurofascin cryptic exon inclusion or gender of mice? Is there assessment using an independent behavioral task to confirm this finding? Given the non-normal distribution, a standard t-test is not appropriate for this statistical comparison.

The complete loss of paranodal junctions has been confirmed in at least two cKO mice per Cre driver line (we used three Cre driver lines: Dhh-Cre, P0-Cre and Cnp-Cre), and the paranodal phenotype has a 100% penetrance. We have included more mice in the rotarod analysis for a total of 10 mice per genotype. The cKO data points are now more evenly distributed (revised Figure 2H), and the normality of the data point distribution is not rejected by the Shapiro-Wilk test (p = 0.1513 for 24 rpm and p = 0.1949 for 32 rpm). The majority of the WT mice could remain on the rotating rod until the cutoff time of 90 sec and indeed the normality of the WT distribution is rejected by the Shapiro-Wilk test. Therefore, we have switched to the Mann-Whitney statistical analysis and now present the data in box-and-whisker plots. The motor deficit of the cKO mice is now confirmed with a larger group of mice and by using the Mann-Whitney test (p = 0.0115 for 24 rpm and p = 0.0036 for 32 rpm). We have revised the manuscript and source data file accordingly.